



# DeLiAn – a growing collection of depolarization ratio, lidar ratio and Ångström exponent for different aerosol types and mixtures from ground-based lidar observations

Athena Augusta Floutsi[1], Holger Baars[1], Ronny Engelmann[1], Dietrich Althausen[1], Albert Ansmann[1], Stephanie Bohlmann[1,a], Birgit Heese[1], Julian Hofer[1], Thomas Kanitz[1,b], Moritz Haarig[1], Kevin Ohneiser[1], Martin Radenz[1], Patric Seifert[1], Annett Skupin[1], Zhenping Yin[1, 2], Sabur F. Abdullaev[3], Mika Komppula[4], Maria Filioglou[4], Elina Giannakaki[4, 5], Iwona S. Stachlewska[6], Lucja Janicka[6], Daniele Bortoli[7], Eleni Marinou[8], Vassilis Amiridis[8], Anna Gialitaki[8, 9, c], Rodanthi-Elisavet Mamouri[10, 11], Boris Barja[12], and Ulla Wandinger[1]

[1]Leibniz Institute for Tropospheric Research (TROPOS), Leipzig, Germany
[2]School of Remote Sensing and Information Engineering, Wuhan University, Wuhan, China
[3]Physical Technical Institute of the National Academy of Sciences of Tajikistan, Dushanbe, Tajikistan
[4]Finnish Meteorological Institute, Kuopio, Finland
[5]Department of Environmental Physics and Meteorology, University of Athens, Athens, Greece
[6]Faculty of Physics, University of Warsaw, Warsaw, Poland
[7]Évora University, Institute for Earth Sciences, Évora, Portugal
[8]IAASARS, National Observatory of Athens, Athens, Greece
[9]Laboratory of Atmospheric Physics, Physics Department, Aristotle University of Thessaloniki, Thessaloniki, Greece
[10]ERATOSTHENES Centre of Excellence, Limassol, Cyprus
[11]Cyprus University of Technology, Department of Civil Engineering and Geomatics, Cyprus
[12]Atmospheric Research Laboratory, University of Magallanes, Punta Arenas, Chile
[a]now at: Finnish Meteorological Institute, Helsinki, Finland
[b]now at: European Space Agency, ESTEC, Noordwijk, the Netherlands
[c]now at: School of Physics and Astronomy, Earth Observation Science Group, University of Leicester, Leicester, UK

**Correspondence:** Athena Augusta Floutsi (floutsi@tropos.de)

**Abstract.** This paper presents a collection of lidar-derived aerosol intensive optical properties for several aerosol types, namely the particle linear depolarization ratio, the extinction-to-backscatter ratio (lidar ratio) and the Ångström exponent. The data collection, named DeLiAn, is based on globally distributed, long-term, ground-based, multiwavelength, Raman and polarisation lidar measurements, conducted mainly with lidars that have been developed at the Leibniz Institute for Tropospheric Research.

The intensive optical properties are presented at two wavelengths, 355 and 532 nm, for 13 aerosol categories. The categories cover the basic aerosol types (i.e., marine, pollution, continental European background, volcanic ash, smoke, mineral dust) as well as the most frequently observed mixtures they form. This extensive collection also incorporates more peculiar aerosol categories, including dried marine aerosol that, compared to marine aerosol, exhibits a significantly enhanced depolarization ratio (up to 15%). Besides Saharan dust, additional mineral dust types related to their source region were identified due to their

lower lidar ratios (Central Asian and Middle Eastern dust). In addition, extreme wildfire events (such as in north America and Australia) emitted smoke into the stratosphere showing significant different optical properties, i.e., high depolarization values





(up to 25%), compared to tropospheric smoke. The data collection reflects and underlines the variety of aerosol mixtures in the atmosphere and can be used for the development of aerosol typing schemes. The paper contains the currently most comprehensive overview of optical properties from aerosol lidar measurements and, therefore, provides a solid basis for future

aerosol retrievals in the frame of both spaceborne and ground-based lidars. Furthermore, DeLiAn can assist the efforts for harmonization of satellite records of aerosol properties performed at different wavelengths.

## 1   Introduction

Aerosol typing is intertwined with the scientific efforts for the quantification of the direct and indirect aerosol radiative effects, the identification of the main aerosol sources and the improvement of the aerosol measurements, retrievals and models. All these

efforts have a common baseline: the reduction of the aerosol-induced uncertainties in the Earth's radiative budget (Boucher et al., 2013). The uncertainties are directly linked to the physico-chemical properties and the spatiotemporal variability of the aerosol particles.

The climate response to aerosols is not only type- but also altitude-dependent and, therefore, the vertical aerosol distribution is a key factor for the evaluation of the direct aerosol radiative effect (Hansen et al., 1997). Lidars are the only instru-

ments that can accurately characterize complex aerosol mixtures as they provide information on the optical and microphysical properties of different aerosol types along with their vertical distribution (Ansmann and Müller, 2005). Lidar-derived intensive optical parameters can be used effectively for aerosol-typing purposes because, in contrast to extensive properties, they are concentration-independent. The extinction-to-backscatter ratio (lidar ratio), the particle linear depolarization ratio and the Ångström exponent (backscatter- and extinction-related) reveal information about the size, shape and absorption efficiency of

the aerosol particles. The combination of different lidar-derived intensive optical properties is proven to be a sophisticated way for the classification of the different aerosol types and their mixtures (e.g., Sasano and Browell, 1989; Sugimoto et al., 2002; Ansmann et al., 2002; Müller et al., 2002, 2003, 2005; Mattis et al., 2002b, 2004; Tesche et al., 2009b, 2011a; Groß et al., 2011, 2013; Weinzierl et al., 2011; Burton et al., 2012; Papagiannopoulos et al., 2018; Nicolae et al., 2018). Especially the combination of the particle linear depolarization ratio and the lidar ratio is highly effective for classification purposes, since

those parameters exhibit the highest discrimination power among the lidar-derived intensive optical parameters (Burton et al., 2012).

Due to the high capabilities of the lidar instruments with respect to aerosol monitoring and characterization, several lidar networks have emerged around the globe in the last decades. The Micro-Pulse Lidar Network (MPLNET) of the National Aeronautics and Space Administration (NASA) has sites mainly across North America (Welton et al., 2002), the Asian Dust

and Aerosol Lidar Observation Network (AD-Net) is expanding in East Asia (Sugimoto et al., 2014), the Latin-American Lidar Network (LALINET) over Latin America (Antuña-Marrero et al., 2017) and most parts of Europe are covered by the European Aerosol Research Lidar Network (EARLINET; Pappalardo et al., 2014). These networks operate different lidar systems and therefore have different capabilities. For instance, MPLNET is equipped with an elastic-backscatter lidar that operates at 532 nm. AD-Net is mostly equipped with Raman lidars operational mainly at 532 and 1064 nm. Similarly, EARLINET stations





are mostly equipped with multiwavelength (355, 532 and 1064 nm) Raman lidars. Elastic lidar systems have limited capabilities (e.g., with respect to aerosol typing) compared to Raman lidar systems, since two physical quantities, the particle backscatter and extinction coefficients, need to be determined by one measured quantity, i.e., the elastic lidar return (Ansmann and Müller, 2005). On the other hand, with the Raman lidar method (Ansmann et al., 1992) the backscatter and extinction coefficients can be determined independently. In addition, spectral information provided by multiwavelength lidars advance aerosol typing and

can be used to derive microphysical particle properties (via inverse modelling).

    Since the 90s, the Leibniz Institute for Tropospheric Research (TROPOS) has conducted extensive research on the topic of lidar and aerosols. Two complex lidar systems named MARTHA and BERTHA have been developed using several different lidar-specific techniques (Raman, polarization, multi-wavelength, high-spectral-resolution, etc.) throughout the years. The mobile container-based BERTHA (Althausen et al., 2000) was deployed at several field campaigns since the end of the 1990's

(e.g., LACE98 (Wendisch et al., 2002), ACE–2 (Ansmann et al., 2002), INDOEX (Ansmann et al., 2000), COPS (Herold et al., 2011), SAMUM 1&2 (Ansmann et al., 2011), SALTRACE (Haarig et al., 2017a)), while the lab-based MARTHA (Mattis et al., 2002a) was used for EARLINET observations (Mattis et al., 2004, 2008) and testing of new methodologies (e.g., Mattis et al., 2002a; Schmidt et al., 2013; Jimenez et al., 2020a). In parallel, novel data retrieval techniques have been developed and steadily improved, which are meanwhile state-of-the-art for active aerosol profiling (Ansmann and Müller, 2005), e.g., inversion tech-

niques (Müller et al., 1998), separation of aerosol components with polarization (POLIPHON, Mamouri and Ansmann, 2016), automatic and unsupervised data retrievals (e.g., Baars et al., 2008, 2016; D'Amico et al., 2015; Baars and Yin, 2020; Yin and Baars, 2021). The intensive work on the inversion technique methodology (e.g., Müller et al., 1999a, b, 2000, 2011) led finally to the conclusion that 3+2 lidars (backscatter coefficient at three wavelengths, extinction coefficient at two wavelengths) are needed as ideal setup for remote sensing of aerosol microphysical properties – a conclusion which is meanwhile also ap-

plied in EARLINET and ACTRIS (Aerosol, Clouds and Trace Gases Research Infrastructure). The introduction and intense use of the polarization technique led to the possibilities to separate dust and non-dust components (Tesche et al., 2011a), fine and coarse mode of dust (Mamouri and Ansmann, 2014) and even to the retrieval of CCN (Cloud Condensation Nuclei) and INP (Ice Nucleating Particles) properties (Mamouri and Ansmann, 2015, 2016) – important parameters to investigate aerosol-cloud-interactions. Also these techniques are meanwhile widely standardized applied in ACTRIS and beyond. Based on the

experience and expertise gathered in the 90s and with the upcoming need for automized observations, in 2002, the first portable, remotely controlled, multiwavelength Raman polarization lidar system (Polly) was developed at TROPOS (Althausen et al., 2009; Engelmann et al., 2016). Since then, more than a dozen lidars of Polly/Polly$^{XT}$ type (more details in Sect. 2.1; Engelmann et al., 2016) have been constructed and are operating within the framework of a voluntary, scientific network called PollyNET (Baars et al., 2016). The mobility of Polly$^{XT}$ systems as well as their automated and continuous 24/7 observational

capabilities make them ideal for deployment in remote places during measurement campaigns (e.g., the Amazon (Baars et al., 2012), China (Hänel et al., 2012; Heese et al., 2016), South Africa (Giannakaki et al., 2016), India (Komppula et al., 2012), and more recently in Cyprus (Ansmann et al., 2019), the Arctic (Engelmann et al., 2021) and Punta Arenas, Chile (Radenz et al., 2021a)) and on research vessels, such as Sonne, Meteor (Rittmeister et al., 2017) and Polarstern (Kanitz et al., 2013a; Bohlmann et al., 2018).



Lidars have also been successfully deployed in space, aiming to contribute to the scientific efforts for atmospheric measurements on a global scale. For a remarkable duration of 16 years, CALIOP (Cloud-Aerosol Lidar with Orthogonal Polarization) on board NASA's CALIPSO (Cloud-Aerosol Lidar and Infrared Pathfinder Satellite Observations) has measured vertical profiles of attenuated backscatter at visible and near-infrared wavelengths, along with depolarization in the visible channel (Winker et al., 2009). However, as an elastic-backscatter lidar, CALIOP is not able to perform direct extinction measurements. To enable

the retrieval of the backscatter and extinction coefficients from the attenuated backscatter signals, the lidar ratio needs to be assumed. Since the lidar ratio depends on the aerosol types present in the atmosphere, an aerosol typing scheme was developed for CALIPSO (Omar et al., 2005, 2009; Kim et al., 2018).

In 2018, the European Space Agency (ESA) launched the wind lidar mission Aeolus (Stoffelen et al., 2005). The satellite is equipped with a 355-nm high-spectral-resolution lidar (HSRL), the Atmospheric Laser Doppler Instrument (ALADIN).

ALADIN is the first wind lidar in space and its HSRL capabilities provide also the first direct-measured extinction profiles and aerosol optical properties from space, as a spin-off product (Flament et al., 2021), as already successfully demonstrated in Baars et al. (2021). This is a great step towards the harmonization of multiple satellite instruments, especially with view on the upcoming Cloud, Aerosol and Radiation Explorer (EarthCARE) joint mission of ESA and the Japanese Aerospace Exploration Agency (JAXA), scheduled for launch in 2023.

EarthCARE's payload consists of four instruments: an ATmospheric LIDar (ATLID), a Cloud Profiling Radar (CPR), a Multi-Spectral Imager (MSI) and a Broad-Band Radiometer (BBR) (Illingworth et al., 2015). ATLID is a 355-nm HSRL that will provide direct cloud and aerosol profile measurements of backscatter and extinction coefficients. Furthermore, ATLID is able to measure the depolarization ratio of the atmospheric particles – an ideal parameter for aerosol typing – as well as ice particle characteristics (Illingworth et al., 2015; do Carmo et al., 2021). The primary goal of EarthCARE is radiative closure,

which is aimed to be achieved in a synergistic approach from the two active and two passive instruments. One key element for this goal is a proper aerosol typing scheme, to calculate the aerosol's radiative properties. For this purpose, the Hybrid End-To-End Aerosol Classification (HETEAC) model has been developed (Wandinger et al. 2016a, Wandinger et al. in preparation). As the name indicates, the HETEAC model delivers the required theoretical description of aerosol microphysics that is consistent with experimentally derived optical properties (hybrid approach) to close the loop from observations and aerosol microphysics

to radiative properties (end-to-end approach).

It is evident that global, vertically resolved observations from ground-based and spaceborne lidars need to be harmonized (i.e., spectral harmonization). The harmonization of, e.g., lidar-derived intensive optical properties would improve the consistency of the lidar data obtained by different lidar systems, and would allow for comprehensive studies on the statistical relations between those properties. Harmonized datasets of intensive optical parameters would not only lead to the creation of robust

aerosol typing algorithms, but would also improve already existing ones.

Given the need for development and improvement of aerosol typing schemes, such as HETEAC, and data harmonization among lidar networks (i.e., MPLNET, AD-Net, LALINET, EARLINET, PollyNET) and satellites (e.g., CALIPSO, Earth-CARE), in this paper, we present an experimental data collection of aerosol-type-dependent optical properties. The optical properties are the particle linear depolarization ratio, the lidar ratio and the Ångström exponent and, hence, the data collection





is named DeLiAn. The optical properties have been obtained either by lidar systems that have been developed at TROPOS (such as MARTHA, BERTHA, Polly$^{\text{XT}}$; more information in Sec.2) or by other accompanying lidar systems (explicitly mentioned in Sec.2) during different field campaigns and at different locations throughout many years (see Tab. 1 and Fig.1). In addition to the well-known aerosol types (such as marine, dust, pollution, etc.), DeLiAn features new findings and aerosol types (e.g., Central Asian dust (Hofer et al., 2020), dried marine (Haarig et al., 2017b), stratospheric smoke (Haarig et al., 2018; Ohneiser

et al., 2020)) that were identified during recent measurement campaigns. The aim is to provide additional knowledge on the intensive aerosol properties for different aerosol types and mixtures, which in turn can be used to constrain and enhance aerosol retrievals.

In the following section, we briefly present the lidar systems used, information about the data handling, and we provide an overview of the locations where the relevant lidar systems have been operated. In Section 3, the collection of the lidar-

derived intensive optical parameters (DeLiAn) is presented and discussed in detail with respect to the different aerosol types and mixtures. The statistical analysis performed is also presented in the same section along with comparisons between the optical properties used for the CALIPSO aerosol typing scheme and the respective ground-based ones that were obtained from the present study. The conclusions and an outlook finalize the paper.

## 2    Instruments, data analysis and sources

### 2.1    Lidar systems

#### 2.1.1    MARTHA

The Multiwavelength Aerosol Raman Lidar for Temperature, Humidity, and Aerosol profiling is a lab-based lidar of TROPOS that has been used not only for acquiring cloud and aerosol measurements but also for testing new methodologies (Mattis et al., 2002a; Schmidt et al., 2013; Jimenez et al., 2019, 2020a). MARTHA's powerful laser together with its large prime mirror

(80 cm in diameter) makes it ideal for tropospheric and stratospheric aerosol observations, and recently has been upgraded to a dual-receiver-field-of-view lidar (RFOV; Jimenez et al., 2020b). The operational setup has been steadily changed but covers at least $3\beta + 2\alpha + \delta$ (three backscatter coefficients at 355, 532 and 1064 nm, two extinction coefficients at 355, 532 nm and one depolarization ratio at 532 nm). MARTHA is part of EARLINET/ACTRIS (Pappalardo et al., 2014) but not yet automatized.

#### 2.1.2    BERTHA

The Backscatter Extinction lidar-Ratio Temperature Humidity profiling Apparatus (BERTHA) is the oldest mobile, container-based multiwavelength polarization Raman lidar of TROPOS (Althausen et al., 2000). For 25 years, BERTHA has been providing extensive and intensive aerosol optical properties and has been deployed at numerous field campaigns (most notably in both the SAMUM and SALTRACE field experiments, more details in Sec. 2.2). Manual operation is needed for this lidar. A distinctive feature of this lidar system is that since 2012, it enables simultaneous measurements of the depolarization ratio

at three wavelengths (355, 532 and 1064 nm). In 2015 the setup was extended by the first measurements of the extinction





coefficient at 1064 nm, leading to a $3\beta + 3\alpha + 3\delta$ setup (three backscatter coefficients, three extinction coefficients and three depolarization ratios). The system includes a water vapor and a HSRL channel (407 and 532 nm, respectively). For a more detailed description of the latest setup of the lidar system readers may refer to Haarig et al. (2016, 2017a).

### 2.1.3 Polly$^{\mathrm{XT}}$ lidar systems

Since the first Polly system (Althausen et al., 2009) was assembled in 2002, more than a dozen lidars of Polly/Polly$^{\mathrm{XT}}$ type have been constructed at TROPOS, with continuous upgrading efforts (Engelmann et al., 2016), and deployed either permanently or temporarily at measurement campaigns and research vessels (for visualization see the online map at polly.tropos.de, last access: 18 October 2022). Some Polly$^{\mathrm{XT}}$ lidars are part of EARLINET/ACTRIS (Pappalardo et al., 2014). Polly$^{\mathrm{XT}}$ Raman lidars emit light at three different wavelengths, 355, 532 and 1064 nm. The lidars are nowadays equipped with a receiver that

consists of twelve or more channels, which enable measurements of the elastically (355, 532, 1064 nm) and Raman-scattered light (387, 607 nm for aerosols and 407 nm for water vapour) and the depolarization state of the backscatter light (at 355 and 532 nm). The near-range telescope allows the detection of scattered light (at 355, 387, 532 and 607 nm) from an altitude of around 60–80 m above ground level (AGL). The uppermost detection height for the vertical profiles is around 20 km AGL. Data from all channels are acquired with a vertical resolution of 7.5 m and a temporal resolution of 30 s (for more details refer

to Engelmann et al., 2016).

This setup allows the determination of extensive and intensive aerosol optical properties, which are important quantities for the monitoring and characterization of the aerosol. The determination of the backscatter coefficient (355, 532 and 1064 nm) and the extinction coefficient (355 and 532 nm) leads to the lidar ratio and Ångström exponent (Baars et al., 2016). Another intensive optical property, the particle linear depolarization ratio, defined as the cross-polarized-to-co-polarized backscatter

ratio (orthogonal and parallel planes of polarization to the plane of linear polarization of the transmitted laser pulses, respectively), is also determined (Baars et al., 2016). Quality assurance procedures are a key aspect for lidars (e.g., Bravo-Aranda et al. 2016; Wandinger et al. 2016b; Freudenthaler 2016; Belegante et al. 2018; Freudenthaler et al. 2018) and the Polly$^{\mathrm{XT}}$ lidar systems and data processing follows the EARLINET standards even when the lidars are operated at non-stationary sites (e.g., on research vessels). Near-real-time (NRT) quicklooks can be found at polly.tropos.de (last access: 18 October 2022).

### 2.2 Lidar data analysis, locations and sources

The data collection presented here comprises various measurements from several locations, an overview is given in Fig. 1. We have considered data that are already published (layer- and observational-mean values) and added additionally analyzed data for specific aerosol types. The evaluation of the published lidar data has been performed by the authors of the corresponding papers. The full list of the respective references can be found in Tab. 1, while a brief description of the major field campaigns

mentioned in the same table is provided below.

Table 1 provides an overview of all the aerosol types that were considered during the creation of DeLiAn. It corroborates previous findings and provides new insights regarding aerosol types based on recent measurement campaigns and studies.



**Table 1.** Overview of the lidar-derived optical properties of different aerosol types. The lidar ratio $S$ is expressed in sr and the particle linear depolarization $\delta$ in % along with the respective errors. References for each category are given in the right column. Measurements conducted in a field campaign are indicated with a number at the reference and explained in the footnote of the table, during a Polarstern or Meteor cruise with a bullet symbol (•), while the rest are from PollyNET/EARLINET stations. Measurements that were conducted with a non-TROPOS lidar, are accompanied with a star symbol (⋆).

| Aerosol type | $S_{355}$ | $S_{532}$ | $\delta_{355}$ | $\delta_{532}$ | Reference |
|---|---|---|---|---|---|
| Ash | $51 \pm 7.5$ | $48 \pm 7.5$ | $36 \pm 2.3$ | - | Groß et al. (2012)⋆, Sicard et al. (2012), Kanitz (2012)• |
| Saharan dust | $53.5 \pm 7.7$ | $53.1 \pm 7.9$ | $24.4 \pm 2.5$ | $28 \pm 1.3$ | Groß et al. (2011)⋆,[2], Preißler et al. (2011), Kanitz et al. (2013a)•, Baars et al. (2016), Rittmeister et al. (2017)•, Kaduk (2017)[6], Haarig et al. (2017a)[4], Urbanneck (2018)[7], Bohlmann et al. (2018)•, Szczepanik et al. (2021), Haarig et al. (2022)[4] |
| Central Asian dust | $43.4 \pm 1.9$ | $37.7 \pm 2.1$ | $22.8 \pm 0.8$ | $32.5 \pm 0.7$ | Hofer et al. (2020)[5] |
| Middle Eastern dust | $39.5 \pm 6$ | $37.4 \pm 5.3$ | $24.2 \pm 2.3$ | $28.4 \pm 1.6$ | Müller et al. (2007)[1], Kaduk (2017)[6], Urbanneck (2018)[7], Filioglou et al. (2020) |
| Smoke | $68.2 \pm 7.4$ | $71.8 \pm 11.1$ | $2.7 \pm 1.3$ | $2.9 \pm 0.6$ | Müller et al. (2007), Baars (2011)[3], Tesche (2011)[2], Pereira et al. (2014), Giannakaki et al. (2016)[3], Janicka et al. (2016), Haarig et al. (2018), Floutsi et al. (2021)[8], Ohneiser et al. (2021)[9],• |
| Stratospheric smoke | $67.5 \pm 19.3$ | $93.8 \pm 18.1$ | $22.6 \pm 4$ | $17.9 \pm 1.7$ | Haarig et al. (2018), Ohneiser et al. (2020)[8] |
| Dust and smoke | $72.1 \pm 7.7$ | $56.3 \pm 6.5$ | $15.7 \pm 2$ | $18.9 \pm 1.4$ | Groß et al. (2011)⋆,[2], Tesche (2011)[2], Kanitz et al. (2013a)⋆,•, Giannakaki et al. (2016)[3], Kanitz et al. (2014b)•, Kaduk (2017)[6], Rittmeister et al. (2017)• |
| Pollution | $51.1 \pm 8.7$ | $47.4 \pm 7.4$ | $1.1 \pm 0.3$ | $2.8 \pm 1$ | Ansmann et al. (2005), Müller et al. (2007), Tesche et al. (2007), Komppula et al. (2012)[3], Preißler et al. (2013), Hänel et al. (2012)[3], Giannakaki et al. (2016)[3], Heese et al. (2017), Kaduk (2017)[6], this study (Leipzig) |
| Dust and pollution | $48.5 \pm 9.2$ | $46.4 \pm 8$ | $15.7 \pm 1.7$ | $17.7 \pm 2.5$ | Leipzig, Germany⋆, Preißler et al. (2013), Janicka et al. (2016), Kaduk (2017)[6], Rittmeister et al. (2017) |
| Dried marine | $28 \pm 6.6$ | $26.9 \pm 10.6$ | $7.5 \pm 1.7$ | $8.3 \pm 1.1$ | Haarig et al. (2017b)[4], Bohlmann et al. (2018)• |
| Clean marine | $22.4 \pm 5.6$ | $21.9 \pm 13.4$ | $1.3 \pm 0.3$ | $1.4 \pm 0.3$ | Groß et al. (2011)⋆,[2], Kaduk (2017)[6], Bohlmann et al. (2018)•, Rittmeister et al. (2017)• |
| Dust and marine | $39.4 \pm 5.6$ | $32 \pm 7.8$ | $14 \pm 1.5$ | $14.7 \pm 1.1$ | Groß et al. (2011)⋆, Kaduk (2017)[6], Bohlmann et al. (2018)• Rittmeister et al. (2017)• |
| Central European background | $57 \pm 4.7$ | $56.2 \pm 8.3$ | $3.4 \pm 1.8$ | $3.2 \pm 0.1$ | Leipzig, Germany, Müller et al. (2007), this study (Leipzig) |

[1]INDOEX, [2]SAMUM, [3]EUCAARI, [4]SALTRACE, [5]CADEX, [6]BACCHUS, [7]CyCARE/A-LIFE, [8]DACAPO-PESO, [9]MOSAiC,





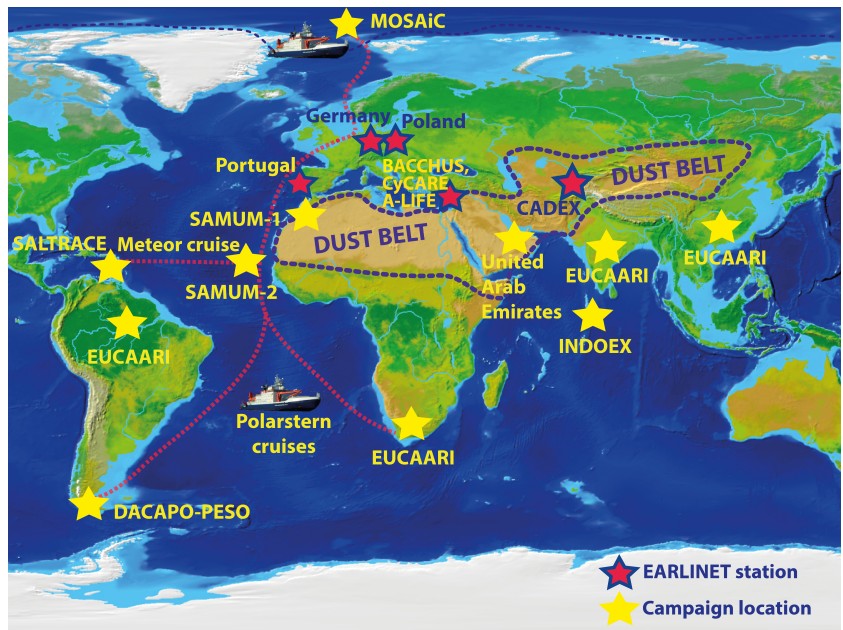

**Figure 1.** Locations of measurement campaigns (yellow stars) or permanent EARLINET stations (red/blue stars) from which data for the current study was used. Map source: primap.

Furthermore, the data collection is regularly updated aiming to provide a comprehensive up-to-date collection of the optical properties of the different aerosol types at the typical lidar wavelengths.

The portable TROPOS lidar systems BERTHA and Polly$^{XT}$ have been deployed at numerous campaigns, research platforms and locations (Fig. 1). In the following, a brief introduction of major field campaigns and research vessels, where TROPOS lidars operated, is presented. Measurements from the field campaigns listed below have contributed greatly to the collection presented here. Campaigns that contributed to the collection with a few measurements only are not listed here. In addition, the locations of all the campaigns relevant for this study and permanent measuring stations are indicated in Fig. 1.

**2.2.1   Research cruises**

Two German Research Vessels (RV), namely Meteor and Polarstern, serve as research platforms where Polly$^{XT}$ lidars have been deployed. The OCEANET-Atmosphere observatory, which includes a Polly$^{XT}$ lidar among other instruments, has been frequently operated aboard Polarstern for the transects from the Northern Hemisphere to the Southern Hemisphere and vice versa (e.g., Kanitz et al., 2013b; Bohlmann et al., 2018; Yin et al., 2019), as well as in the polar regions (e.g., Griesche
et al., 2020; Engelmann et al., 2021). Pure marine aerosol conditions as well as complex mixtures, typically including sea-salt i.e., dust and marine aerosol mixtures, are being observed frequently. Data acquired during the transatlantic ship cruises and MOSAiC (more about that campaign in Sect. 2.2.9) have contributed greatly to the scientific efforts for aerosol monitoring,





**Table 2.** Extinction- ($AE_{355/532}$) and backscatter-realted ($AE_{b355/532}$ and $AE_{b532/1064}$) Ångström exponents for the aerosol categories listed in Tab. 1.

| Aerosol type | $AE_{355/532}$ | $AE_{b355/532}$ | $AE_{b532/1064}$ |
|---|---|---|---|
| Ash | $0.8 \pm 0.6$ | $0.6 \pm 0.4$ | $1 \pm 0.4$ |
| Saharan dust | $0.1 \pm 0.2$ | $0.03 \pm 0.08$ | $0.5 \pm 0.1$ |
| Central Asian dust | $0.2 \pm 0.1$ | $-0.2 \pm 0.03$ | $0.4 \pm 0.01$ |
| Middle Eastern dust | $0.1 \pm 0.1$ | $0.4 \pm 0.2$ | $0.7 \pm 0.2$ |
| Smoke | $1.3 \pm 0.3$ | $1.4 \pm 0.1$ | $1.2 \pm 0.1$ |
| Stratospheric smoke | $-0.3 \pm 0.4$ | $1.2 \pm 0.4$ | $1.2 \pm 0.6$ |
| Dust and smoke | $1.4 \pm 0.2$ | $0.5 \pm 0.1$ | $1 \pm 0.05$ |
| Pollution | $1.8 \pm 1.4$ | $1.2 \pm 0.7$ | $0.9 \pm 0.5$ |
| Dust and pollution | $0.7 \pm 0.4$ | $0.3 \pm 0.1$ | $0.9 \pm 0.1$ |
| Dried marine | $1.1 \pm 1.3$ | $0.6 \pm 0.04$ | $-0.07 \pm 0.07$ |
| Clean marine | $0.7 \pm 1.3$ | $0.8 \pm 0.1$ | $0.5 \pm 0.1$ |
| Dust and marine | $0.5 \pm 0.5$ | $0.3 \pm 0.1$ | $0.6 \pm 0.1$ |
| Central European background | $1.5 \pm 0.2$ | $1.4 \pm 0.2$ | $1.2 \pm 0.2$ |

**Table 3.** Overview of RV cruises that are relevant for this study. The RV, cruise identification sequence and the corresponding study are indicated.

| RV | Cruise ID | Reference |
|---|---|---|
| Polarstern | ANT-XXVI/1, ANT-XXVI/4 and ANT-XXVII/1 | Kanitz et al. (2013a) |
| Polarstern | PS95 and PS98 | Bohlmann et al. (2018) |
| Meteor | M96 | Rittmeister et al. (2017) |
| Polarstern | MOSAiC20192020 | Ohneiser et al. (2021) |

by reaching places where no permanent stations can be established. Tab. 3 shows the identification sequences for the cruises relevant to this study.

### 2.2.2 SAMUM

The Saharan Mineral Dust Experiment (SAMUM) was focused on the investigation of the relationship between chemical composition, shape morphology, size distribution and optical effects of the dust particles originating from the Saharan desert (Ansmann et al., 2011). Two field campaigns were conducted in southern Morocco in 2006 and in Cape Verde in 2008 (SAMUM–1 and SAMUM–2, respectively). The combination of the ground-based multiwavelength Raman lidar BERTHA, two lidar sys-





tems from the University of Munich (the Portable Lidar System POLIS (Groß et al., 2008) and MULIS) and the airborne HSRL measurements (Falcon-20 research aircraft of the German Aerospace Center, DLR) allowed the profiling of pure dust optical properties as well as mixed aerosol plumes and provided a unique dataset that has been extensively used for radiation closure studies, development of appropriate dust parameterizations for large-scale and regional weather and climate models and in-situ comparison studies (Tesche et al., 2009a, b, 2011a, b).

### 205  2.2.3  EUCAARI campaigns

The European Integrated Project on Aerosol, Cloud, Climate, Air Quality Interactions project (EUCAARI; Kulmala et al., 2011) was a multidisciplinary project focused on the interactions between climate and air pollution. EUCAARI lasted for three years (01.01.2007–31.12.2010) and resulted in comprehensive datasets of aerosol properties from Europe and from four non-European countries (China, India, Brazil and South Africa). In the present study, measurements from all the aforementioned

countries are included with a specific focus on measurements that were conducted in two EUCAARI campaigns in Amazonia and South Africa. A Polly$^{\text{XT}}$ lidar was deployed for the first time near Manaus, Brazil from January to November 2008. The long-term lidar observations obtained during that campaign have advanced our knowledge on the vertical aerosol distribution of Saharan dust, biomass-burning aerosol (BBA) and their mixtures, which get advected from Africa, and prevail the aerosol conditions of Amazonia (Baars, 2011). In addition, a clear distinction between the prevailing aerosol conditions during the

wet (January – June) and dry season (July – November) was achieved (Baars, 2011). A Polly$^{\text{XT}}$ lidar was also operated in Elandsfontein, South Africa from 30 January 2010 to 31 January 2011. During that period biomass-burning aerosol from natural phenomena (lightning) and human-induced activities as well urban and industrial aerosol of anthropogenic origin were observed (Giannakaki et al., 2016). Measurements from the EUCAARI stations in China (Hänel et al., 2012) and India (Komppula et al., 2012) were heavily influenced by pollution-related aerosol. Complex mixtures of aged desert dust, biomass

burning smoke and industrial pollution were frequently observed.

### 2.2.4  SALTRACE

The Saharan Aerosol Long-range Transport and Aerosol-Cloud interaction Experiment (SALTRACE) was conducted from spring 2013 to summer 2014 at Barbados (Weinzierl et al., 2017). The campaign involved ground-based and airborne in-situ and remote-sensing observations. The main goal of SALTRACE was the characterization of Saharan dust particles after long-

range transport (Groß et al., 2015; Haarig et al., 2017a) and its interaction with clouds (Haarig et al., 2019), as a follow-on to the SAMUM field campaign. TROPOS contributed with the deployment of the multiwavelength Raman lidar BERTHA (see Sect. 2) during the dusty summer conditions and the rather clear marine conditions in winter (Haarig et al., 2017a, b).

### 2.2.5  CADEX

The Central Asian Dust Experiment (CADEX; Hofer et al., 2017, 2020) was focused on long-term observations of the op-

tical and microphysical properties of Central Asian mineral dust. It was the first time that ground-based lidar observations





(performed with a Polly$^{\text{XT}}$ system at Dushanbe, Tajikistan) were conducted in Central Asia. CADEX lasted for two years (2014–2016, measurements conducted between 2015 and 2016) and resulted in major findings on optical properties of Central Asian dust, thus, establishing it as a separate aerosol type. In addition, the campaign pointed out the necessity of a permanent ground-based station in the area, which was later on established as part of EARLINET. It is obvious that campaigns aiming at
monitoring a specific aerosol type, in a specific location where it is found in abundance, are very important in the global efforts of aerosol-type standardization.

### 2.2.6 BACCHUS

BACCHUS (Impact of Biogenic versus Anthropogenic emissions on Clouds and Climate: towards a Holistic UnderStanding) was an European collaborative project led by ETH Zurich (https://www.bacchus-env.eu/, last access: 18 October 2022). In the
framework of BACCHUS, a Polly$^{\text{XT}}$ lidar system performed measurements in Nicosia, Cyprus in spring 2015. Pure aerosol types such as dust (Saharan and Middle Eastern), marine, and pollution as well as complex mixtures including smoke were regularly observed (Kaduk, 2017).

### 2.2.7 CyCARE and A-LIFE

The Cyprus Clouds Aerosol and Rain Experiment (CyCARE; Ansmann et al., 2019) took place at Limassol, Cyprus from
October 2016 to March 2018 with main focus on the complex aerosol mixtures, vertical aerosol layering, and their influence on cloud evolution and precipitation processes. Polly$^{\text{XT}}$, part of the LACROS (Leipzig Aerosol and Cloud Remote Observations System, the ground-based remote-sensing supersite of TROPOS; Bühl et al., 2013), was deployed at Limassol in 2017. During the deployment, the A-LIFE (Absorbing aerosol layers in a changing climate: aging, lifetime and dynamics) campaign took place (April 2017), which was led by the University of Vienna. A-LIFE aimed to investigate the properties of absorbing aerosol
and in particular those of mineral dust, black carbon and their mixtures (https://www.a-life.at/, last access: 18 October 2022). In parallel to the A-LIFE campaign, the PRE-TECT campaign took place at the Greek atmospheric observatory of Finokalia in Crete, Greece. The campaign was led by the National Observatory of Athens (NOA) and aimed to improve the desert dust characterization from remote-sensing measurements. For that reason, the Polly$^{\text{XT}}$ lidar system of NOA was deployed at Finokalia.

### 255 2.2.8 DACAPO-PESO

The Dynamics, Aerosol, Clouds, And Precipitation Observations in the Pristine Environment of the Southern Ocean (DACAPO-PESO) field campaign took place in Punta Arenas, Chile and it was focused on the investigation of cloud formation and aerosol-cloud interaction in environments of contrasting aerosol conditions (Radenz et al., 2021a). LACROS (Bühl et al., 2013) was measuring continuously for a period of three years (2018–2021) and with respect to aerosol, significant lofted aerosol layers
were observed occasionally in the troposphere of Punta Arenas (Floutsi et al., 2021). Furthermore, stratospheric smoke originating from the record-breaking Australian bushfires in January 2020 was fully captured by the lidar observations (Ohneiser





et al., 2020). However, the observations confirmed that in the general clean environment of Punta Arenas the aerosol backscatter is on average more than 30% below the mean of the backscatter observed in Europe (Limassol and Leipzig) (Radenz et al., 2021a). In addition, it was found that at Punta Arenas most aerosol is confined in the boundary layer with pristine conditions

dominating aloft (Radenz et al., 2021a).

### 2.2.9   MOSAiC

During the MOSAiC (Multidisciplinary drifting Observatory for the Study of Arctic Climate) expedition, a Polly$^{\text{XT}}$ multiwavelength polarization Raman lidar was operated onboard RV Polarstern from October 2019 to October 2020. For the first time, Polly$^{\text{XT}}$ conducted continuous measurements of aerosols and clouds (up to 30 km altitude) in the central Arctic (Engel-

mann et al., 2021). This unique dataset provided new insights about smoke trapped in the upper troposphere/lower stratosphere (UTLS) of the High Arctic in the winter of 2019–2020 (Ohneiser et al., 2021) and improved our knowledge on aerosol-cloud interactions (Engelmann et al., 2021).

### 2.2.10   EARLINET stations

Measurements from four permanent EARLINET stations have also significantly contributed to this study. The stations are

mainly located in Europe: Leipzig (Germany), Évora (Portugal) and Warsaw (Poland). Pure aerosol types such as smoke, Saharan dust and pollution as well as their complex mixtures are frequently observed above the aforementioned stations. A non-European station located at Dushanbe (Tajikistan; see Sec. 2.2.5) was also considered.

## 3   DeLiAn

### 3.1   Intensive optical properties at 355 and 532 nm

Two intensive aerosol optical properties from the collection of ground-based observations described above (Sec. 2.2), namely the lidar ratio and the particle linear depolarization ratio at 355 nm, are contrasted against each other for several aerosol types and mixtures in Fig. 2. The aforementioned intensive properties exhibit the highest discriminatory power (e.g., as demonstrated in Burton et al., 2012). The dataset used for the conceptualization of the aerosol-typing-related activities of EarthCARE (Wandinger et al., 2016a; Illingworth et al., 2015) has been merged with the present dataset and is shown as semi-transparent

symbols. Additionally to the intensive optical properties at 355 nm, in Fig. 3 we present the values for the same aerosol types at 532 nm. Furthermore, we show the wavelength dependence, i.e., the Ångström exponent contrasted to the lidar ratio and particle linear depolarization ratio, as presented in Fig. 4. Having the intensive properties at 355 and 532 nm allows not only the use of the spectral dependence for typing purposes, but also facilitates the bridging of datasets obtained at one of these wavelengths only as, e.g., in case of the spaceborne lidars CALIOP (532 nm) and ALADIN and ATLID (355 nm) or in the case

of spaceborne and ground-based lidars (Amiridis et al., 2015).





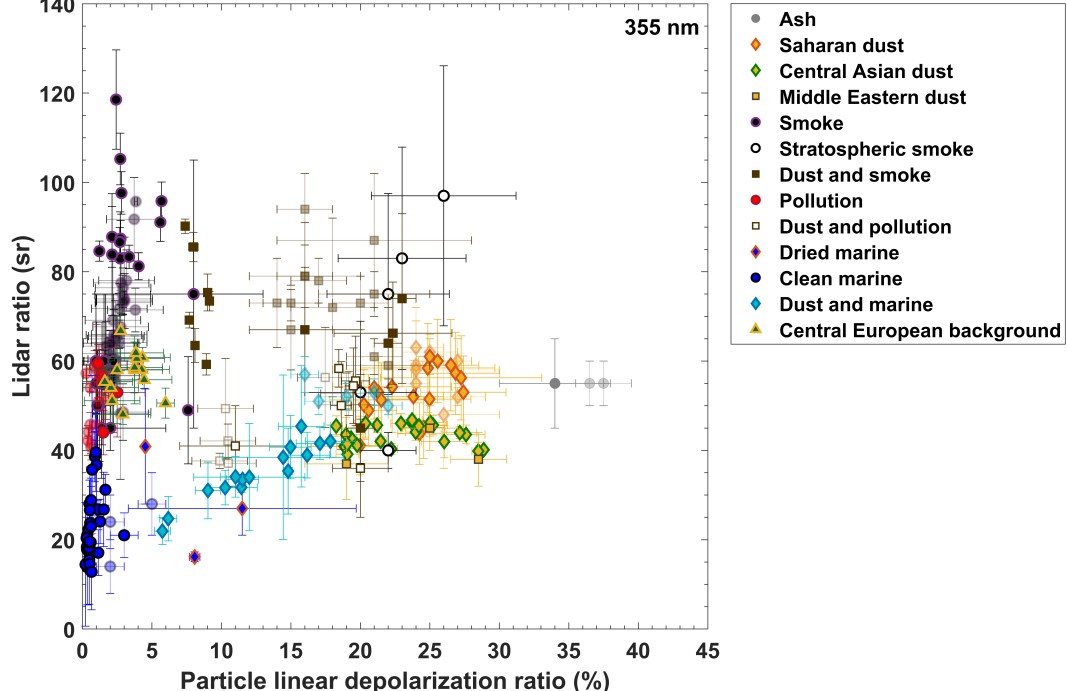

**Figure 2.** Intensive optical properties of different aerosol types, measured at 355 nm. The dataset used for the conceptualization of Earth-CARE's typing scheme is shown with faded markers.

**Ash**

The volcanic ash category contains measurements of the Eyjafjallajökull eruption (April 2010) observed from various locations, including Maisach, Germany (faded grey circles in Fig. 2; Groß et al., 2012) with POLIS, near Bremerhaven, Germany conducted onboard the RV Polarstern (at 532 nm; Kanitz, 2012) and Évora, Portugal (no depolarization information available; Sicard et al., 2012). The mean lidar ratio and particle linear depolarization ratio at 355 and 532 nm and the mean Ångström exponents are reported in Tab. 1 and Tab. 2, respectively.

**Desert dust**

Mineral dust is an important constituent of the atmospheric aerosol load, and in DeLiAn three aerosol categories have been dedicated for this specific aerosol type. Saharan dust has been targeted in many field campaigns and is the first of the three aforementioned categories (orange/red rhombuses in Fig. 2). Saharan dust is frequently advected above Europe, and observations over Portugal (Preißler et al., 2011), Warsaw (Szczepanik et al., 2021) and over Leipzig (Baars et al., 2016) have been included in the data collection. Long-range transported Saharan dust observed over Barbados during SALTRACE in 2013 and 2014 was also considered in the collection (Haarig et al., 2017a, 2022). The island of Cyprus, located in the eastern part



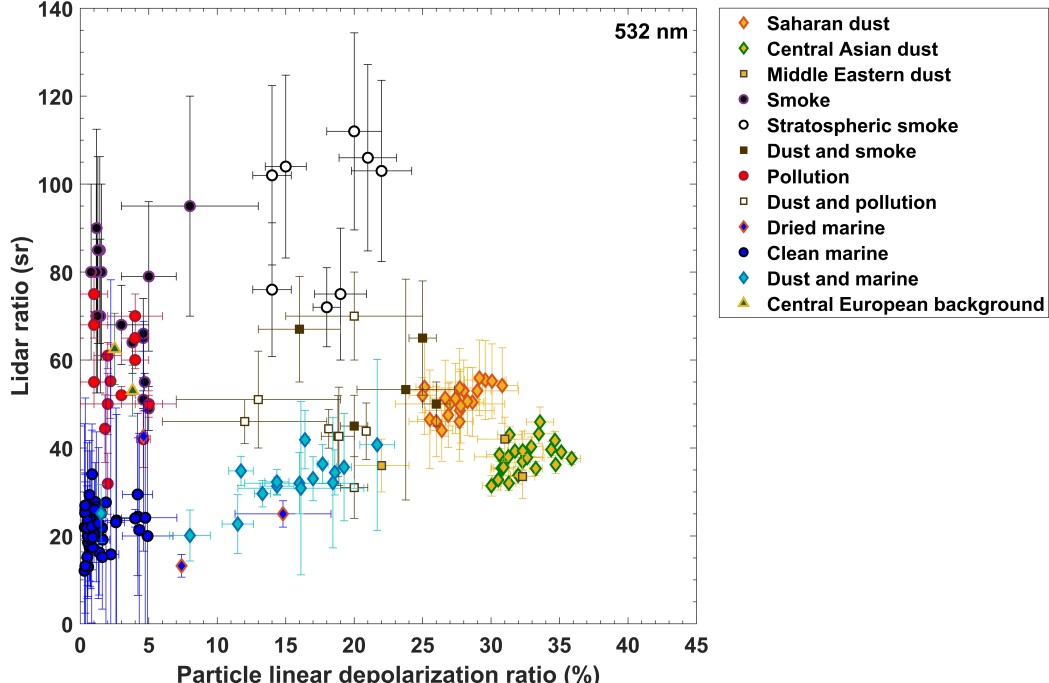

**Figure 3.** Intensive optical properties of different aerosol types, measured at 532 nm.

of the Mediterranean basin, is frequently affected by Saharan dust plumes and, therefore, dust measurements from Limassol

(CyCARE/A-LIFE; Urbanneck, 2018) and from Nicosia (BACCHUS; Kaduk, 2017) were used. Saharan dust has also been observed from onboard the RV Polarstern, over the Atlantic Ocean (faded orange/red rhombuses in Fig. 2; Kanitz et al., 2013a) and near the Canary islands (Bohlmann et al., 2018), as well as from onboard the RV Meteor during a cruise from Guadeloupe to Cape Verde (Rittmeister et al., 2017). Measurements at 355 nm conducted with POLIS during the SAMUM–2 campaign at Cape Verde in 2008 were also considered (faded orange/red rhombuses in Fig. 2; Groß et al., 2011). At the current state

of the experimental data collection, the representative mean lidar ratio and particle linear depolarization values at 355 nm are $53.5 \pm 7.7$ sr and $24.4 \pm 2.5\%$, respectively. Optical information at wavelengths beyond EarthCARE's wavelength (i.e., 355 nm) are presented in Fig. 3 (532 nm) and the Ångström exponents are shown in Fig. 4. At 532 nm, the mean lidar ratio is $53.1 \pm 7.9$ sr and the mean particle linear depolarization ratio is $28 \pm 1.3\%$. Mean values of the extinction-related Ångström exponent are $0.1 \pm 0.2$ and the backscatter-related Ångström exponent values at both 355/532 and 532/1064 nm are around

$0.03 \pm 0.08$ and $0.5 \pm 0.1$, respectively (Tab. 2 and Fig. 4).

In contrast to the findings from the major dust-targeting campaigns, SAMUM–1/2 and SALTRACE, in recent years it was confirmed that Central Asian dust has different optical properties than Saharan dust. Mineral dust that originates from the Central Asian deserts typically exhibits lower lidar ratio values (35–45 sr) than Saharan dust (50–60 sr), as shown by Hofer et al. (2017) on the basis of long-term observations in Tajikistan and, hence, has been introduced to DeLiAn as a separate





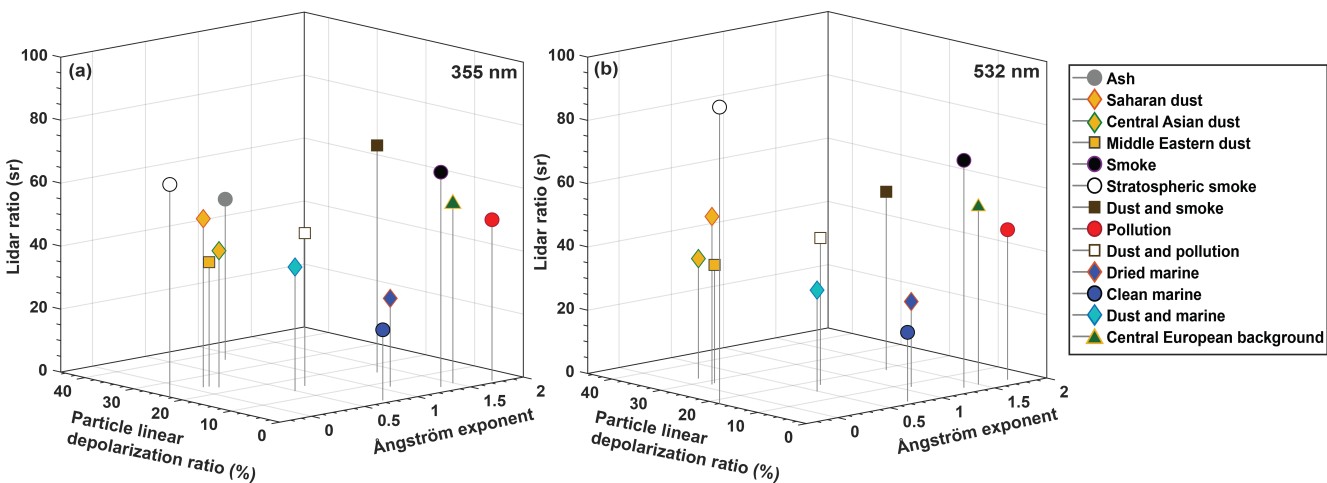

**Figure 4.** Mean lidar ratio and particle linear depolarization ratio at (a) 355 nm and (b) 532 nm versus the mean extinction-related Ångström exponent for the 13 aerosol categories.

aerosol type (green/yellow rhombuses in Fig. 2). The Central Asian dust measurements were conducted in the framework of the CADEX field campaign at Dushanbe, Tajikistan (Hofer et al., 2017, 2020). Mean lidar ratio values for this aerosol category are $43.4 \pm 1.9$ and $37.7 \pm 2.1$ sr at 355 and 532 nm, respectively. The mean particle linear depolarization ratios are $22.8 \pm 0.8\%$ and $32.5 \pm 0.7\%$ at 355 and 532 nm, respectively. Mean values of the extinction-related Ångström exponent are similar to those of Saharan dust $0.2 \pm 0.1$, while the mean backscatter-related Ångström exponent values are $-0.2 \pm 0.03\%$ and $0.4 \pm 0.01\%$ at the wavelength pairs of 355/532 and 532/1064 nm, respectively.

Since the Eastern Mediterranean region has been characterized as a primary climate change "hot spot" (Lelieveld et al., 2012), atmospheric measurements and campaigns have been intensified over the region. Complex mixtures of desert dust, biomass-burning and pollution aerosol are usually encountered. Typically, dust particles observed above that region originate either from the Sahara or the Middle Eastern deserts. Therefore, in addition to the Saharan and Central Asian Dust categories we have introduced a new aerosol type: Middle Eastern dust. At the moment, this aerosol category (brown/yellow squares in Fig. 2) comprises dust originating from Saudi Arabia (Müller et al., 2007) and observed within the framework of the Indian Ocean Experiment (INDOEX; Ramanathan et al., 2001), from the United Arab Emirates (Filioglou et al., 2020) and mineral dust measurements from Cyprus, specifically in Limassol (CyCARE/A-LIFE; Urbanneck, 2018) and in Nicosia (BACCHUS; Kaduk, 2017). On average, the lidar ratios at 355 and 532 nm are $39.5 \pm 6$ sr and $37.4 \pm 5.3$ sr and the particle linear depolarization ratios are $24.2 \pm 2.3\%$ and $28.4 \pm 1.6\%$, respectively. The extinction- and backscatter-related Ångström exponents are $0.1 \pm 0.1$, $0.4 \pm 0.2$ (355/532 nm) and $0.7 \pm 0.2$ (532/1064 nm), respectively.



**Smoke**

The smoke category used for the conceptualization of EarthCARE's classification approach is based on measurements of smoke that were conducted during a EUCAARI campaign in the the Amazon Basin in 2008 (Baars et al., 2012). Measurements of
lidar ratio and especially particle linear depolarization ratio at 355 nm were rare at the time of the campaign (depicted with faded black/purple circles in Fig. 2). Since then, 355 nm lidar measurements have become much more available and, therefore, smoke observations from other locations and fire types (e.g., smoldering or flaming combustion) have been observed too (depicted by black/purple circles in Fig. 2). Smoke observations from another EUCAARI campaign, this time from Elandsfontein, South Africa (Giannakaki et al., 2016) have been included in the data basis. Similarly, smoke measurements performed in the
framework of SAMUM–2 at Cape Verde (Tesche, 2011) were added. Regional smoke and smoke from Australia was observed in the southern tip of South America (Punta Arenas, Chile) during the DACAPO-PESO field campaign (Floutsi et al., 2021). Smoke from Siberian wildfires, which was measured in the Arctic during the MOSAiC campaign, has also been added to the data collection (Ohneiser et al., 2021), which broadens even further the geographical coverage of the observations of that particular aerosol type. Smoke has been also observed above Europe frequently; from boreal forest fires in western Canada at
Leipzig, Germany (Haarig et al., 2018), aged from Siberia/Canada measured at multiple EARLINET stations (Müller et al., 2007), long-range transported smoke from North America above Warsaw, Poland (Janicka et al., 2016, 2017) and fresh, locally-produced smoke above Portugal (Pereira et al., 2014). The 355 nm mean lidar ratio and particle linear depolarization ratio are $68.2 \pm 7.4$ sr and $2.7 \pm 1.3\%$, respectively. At 532 nm, the mean smoke lidar ratio is $71.8 \pm 11.1$ sr and, thus, higher and wider distributed compared to the 355 nm mean lidar ratio. The mean smoke particle linear depolarization ratio at the same wave-
length is $2.9 \pm 0.6\%$. Extinction-related Ångström exponent values are on average around $1.3 \pm 0.3$, and backscatter-related Ångström exponent values are $1.4 \pm 0.1$ and $1.2 \pm 0.1$ for the wavelength pairs of 355/532 and 532/1064 nm.

**Stratospheric smoke**

The different properties of smoke in the troposphere and in the stratosphere resulting from Pyrocumulonimbus convection were not realized before the intense Canadian wildfires of 2017 and were first studied by Haarig et al. (2018) and Ansmann
et al. (2018) and the event's long-term evolution over Europe based on EARLINET measurements by Baars et al. (2019). In contrast to tropospheric smoke, stratospheric smoke is characterized by high depolarization ratios (at both 355 and 532 nm), a feature that was first observed in an elevated aged smoke layer in the upper troposphere, on the eastern seaboard of the United States, with an HSRL onboard the NASA B200 aircraft by Burton et al. (2015). In this data collection, the stratospheric smoke category (depicted by black/white circles in Fig. 2) includes the the aforementioned Canadian wildfires stratospheric smoke
measurements (Haarig et al., 2018) and observations from the record-breaking Australian wildfires of January 2020 measured at Punta Arenas during DACAPO-PESO (Ohneiser et al., 2020). Mean lidar ratios for this aerosol category are $67.5 \pm 19.3$ sr and $93.8 \pm 18.1$ sr at 355 and 532 nm, respectively and, thus, significantly higher, on average, than for tropospheric smoke. Mean particle linear depolarization ratios are $22.6 \pm 4\%$ and $17.9 \pm 1.7\%$. This is a significant finding, as aerosol particles in the stratosphere were usually attributed to volcanic origin (or e.g., generically classified as "stratospheric features" in the version





3 CALIPSO data; Kim et al., 2018). With these new findings it might be possible to distinguish stratospheric aerosol in more

detail and new findings may resurface from historic datasets.

**Pollution**

The pollution category includes measurements that have been conducted in cities of Europe, Asia and Africa. In particular,

the category contains pollution measurements from multiple EARLINET stations in Europe (Müller et al., 2007), including

the urban EARLINET station in Leipzig, from Évora (Preißler et al., 2013), from Nicosia (BACCHUS; Kaduk, 2017), from

Elandsfontein (EUCAARI; Giannakaki et al., 2016), from China (more specifically at Xinken (Ansmann et al., 2005), at

Beijing (Tesche et al., 2007), at Shangdianzi (Hänel et al., 2012) and at Guangzhou (in the framework of the German project

"Megacities–Megachallenges – Informal Dynamics of Global Change"; Heese et al. (2017))) and from India (Gual Pahari- near

New Dehli; Komppula et al., 2012). The mean 355-nm lidar ratio and particle linear depolarization ratio are $51.1 \pm 8.7$ sr and

$1.1 \pm 0.3\%$, respectively. The corresponding properties for the 532 nm are $47.4 \pm 7.4$ sr and $2.8 \pm 1\%$. Ångström exponents are

reported in Tab. 2 and reflect the small size of the specific aerosol type.

**Marine**

The clean marine category (depicted with blue/black circles in Fig. 2) contains measurements of marine particles (i.e., sea

salt) that were conducted mostly onboard a RV or at a coastal station. Measurements conducted during two Polarstern cruises

(PS95 from Bremerhaven, Germany to Cape Town, Republic of South Africa and PS98 from Punta Arenas to Bremerhaven;

Bohlmann et al., 2018), during a Meteor cruise (M96 from Guadeloupe to Cape Verde; Rittmeister et al., 2017), at Nicosia,

Cyprus (Kaduk, 2017) as well as at Cape Verde (faded blue/black circles in Fig. 2; SAMUM–2; Groß et al., 2011) have been

included to the collection. At 355 nm the mean lidar ratio for this aerosol category is $22.4 \pm 5.6$ sr, while the mean particle linear

depolarization ratio is $1.3 \pm 0.3\%$. At 532 nm, the mean lidar ratio is $21.9 \pm 13.4$ sr and the mean particle linear depolarization

ratio is $1.4 \pm 0.3\%$, both indicating a very weak wavelength dependency. Ångström exponent values are on average $0.7 \pm 1.3$

for the extinction-related, $0.8 \pm 0.1$ and $0.5 \pm 0.1$ for the backscatter-related (355/532 and 532/1064 nm, respectively).

Even though the optical properties of aerosol of marine origin do not show variability with respect to the source, when

marine particles are exposed to dry atmospheric conditions, typically with relative humidity lower than 45%, they adopt a

cubic-like shape due to the sodium chloride contained in the sea salt aerosol (Zieger et al., 2017). The cubic-like shape of the

particles causes significantly higher particle linear depolarization ratios (Haarig et al., 2017b; Bohlmann et al., 2018). This fact

was not realized earlier, e.g., in Illingworth et al. (2015) or in aerosol typing schemes such as the one of CALIPSO (Omar et al.,

2005; Kim et al., 2018) and may have led to misclassification of this specific type, e.g., as mixture containing depolarizing

mineral dust particles. Therefore, the introduction of the dried marine aerosol as a separate category was essential. The category

includes measurements conducted at Barbados (2014) during SALTRACE (Haarig et al., 2017b), and above the Atlantic Ocean

with RV Polarstern (Bohlmann et al., 2018) and are always observed on top of the local marine boundary layer. Measurements

of dried marine particles are sparse, leading to only three measurements in this category. Therefore, the statistics presented

here should be interpreted with caution. Dried marine particles exhibit on average lidar ratios of $28 \pm 6.6$ sr and $26.9 \pm 10.6$ sr





and particle linear depolarization ratios of $7.5\pm1.7\%$ and $8.3\pm1.1\%$ at 355 and 532 nm, respectively. The Ångström exponent values are presented in Tab. 2.

**Central European background**

While optically similar to the pollution aerosol category, the "Central European background" aerosol type has been introduced to the collection, given the plethora of the permanent ground-based stations in the indicated geographical area and its resemblance to the more generalized aerosol category of clean continental in the CALIPSO typing scheme. The category (depicted with yellow/green triangles in Fig. 2) includes measurements of the background aerosol load in a typical Central European region (background aerosol measurements performed at Leipzig, and from multiple EARLINET stations, as adapted from Müller et al., 2007). Overall, this aerosol category exhibits slightly higher lidar ratios and particle linear depolarization ratios than pollution (Tab. 1 and 2).

**Mixtures**

Aerosol mixtures of dust particles with smoke, pollution and marine particles have been considered in DeLiAn. Dust and smoke mixtures from Cape Verde (SAMUM–2; Groß et al., 2011) and from measurements above the Atlantic ocean (RV Polarstern; Kanitz et al., 2013a) served as the starting point for the conceptualization of HETEAC (faded brown squares in Fig. 2). Additional measurements from Cape Verde (SAMUM–2; Tesche, 2011), from the Atlantic (conducted while onboard RV Polarstern, near Cape Verde and in the Caribbean; Kanitz et al. (2014b), and while onboard RV Meteor; Rittmeister et al. (2017)), in Elandsfontein (EUCAARI; Giannakaki et al., 2016) and in Nicosia (BACCHUS, Kaduk, 2017) were added to the data collection. The mean lidar ratio and particle linear depolarization ratio for dust-and-smoke mixtures at 355 nm are $72.1\pm7.7$ sr and $15.7\pm2\%$, respectively. The same parameters for the newly-added 532-nm wavelength aree $56.3\pm6.5$ sr and $18.9\pm1.4\%$, respectively. Extinction- and backscatter-related Ångström exponents vary with wavelength, reflecting the versatile particle size range of the mixture (exact values in Tab. 2).

Mixtures of dust and marine particles (cyan/blue rhombuses in Fig. 2) are frequently observed in the lowermost atmosphere, especially in coastal stations such as in Cape Verde (faded cyan/blue rhombuses in Fig. 2; SAMUM–2; Groß et al., 2011) and in Nicosia, Cyprus (Kaduk, 2017). Dust and marine mixtures are also frequently observed with the RV Polarstern (Bohlmann et al., 2018) and RV Meteor (Rittmeister et al., 2017). The 355 nm mean lidar ratio and particle linear depolarization ratio are $39.4\pm5.6$ sr and $14\pm1.5\%$, respectively. Accordingly, $32\pm7.8$ sr and $14.7\pm1.1\%$ for the 532 nm wavelength. The mean extinction-related Ångström exponent is $0.5\pm0.5$ and mean backscatter-related Ångström exponents are $0.3\pm0.1$ and $0.6\pm0.1$ for the wavelength pairs of 355/532 and 532/1064 nm, respectively.

Mixtures of dust and pollution (brown/white squares in Fig. 2) have been observed at Évora (Preißler et al., 2013), Warsaw (Janicka et al., 2016, 2017), Nicosia (Kaduk, 2017) and while onboard the RV Meteor above the Atlantic Ocean (Rittmeister et al., 2017). The mean lidar ratios for this category are $48.5\pm9.2$ sr and $46.4\pm8$ sr, and particle linear depolarization ratios are $15.7\pm1.7\%$ and $17.7\pm2.5\%$ at 355 and 532 nm, respectively. The mean extinction- and backscatter-related (355/532 and 532/1064 nm) Ångström exponents are $0.7\pm0.4$, $0.3\pm0.1$ and $0.9\pm0.1$, respectively.





## 3.2 Statistical analysis of intensive optical properties

The main findings of this study are summarized in Fig. 4, which depicts the mean values of lidar ratio and particle linear depolarization ratio at (a) 355 nm and (b) 532 nm, respectively, against the mean extinction-related Ångström exponent for the different aerosol categories of Tab. 1. Incorporating the Ångström exponent as a third dimension results in more distinctive
aerosol categories (in comparison to Fig. 2 and 3). For instance, the "Central European background" aerosol category can be now clearly distinguished from the pollution category as it exhibits higher extinction-related Ångström exponent values. The respective 2D plots, including the backscatter-related Ångström exponents are shown in Appendix A (Fig. A1 and A2).

A statistical analysis of the intensive optical properties was performed for the 13 different aerosol categories. It should be noted that since the data for each aerosol category have been collected from various sources (see Tab. 1), they naturally exhibit
a variation in the number of data points per aerosol category. In addition, the measurements were performed by lidar systems with different capabilities and, therefore, not all intensive optical parameters were always available for all the observations (e.g., lidar ratio available only at one wavelength, or information on depolarization completely missing). Nevertheless, all available measurements were considered for the creation of DeLiAn and the statistical analysis presented here, and the number of data points used are listed in Tab. B1. The statistics for the lidar ratio and the particle linear depolarization ratio are presented in the
form of boxplots, in Fig. 5 and Fig. 6 for the 355 nm and 532 nm wavelength, respectively. The statistics for all the Ångström exponents are presented in Fig. 7. The minimum and maximum values are indicated by the lower and upper whisker, while the median and mean values by the red lines and rhombuses, respectively. The lower part of each box indicates the 25% percentile and the upper part the 75% percentile. Red crosses represent the outliers (values greater than 1.5 times the interquartile range).

## Comparison with CALIPSO aerosol subtypes

Statistics on the aerosol-type-separated optical properties can be used not only for the development of typing schemes, but can also consolidate already existing aerosol classification schemes, such as the typing scheme of CALIPSO (Omar et al., 2009; Kim et al., 2018). In Fig. 8, the lidar ratio (532 nm) assumed for the different aerosol types in the CALIPSO scheme is contrasted to the observations from the different ground-based lidars (Tab. 1). The CALIPSO lidar ratios for the different
aerosol subtypes are adapted from Tab. 2 of Kim et al. (2018) (version 4). The version 4 stratospheric aerosol subtypes of "Polar stratospheric aerosol" and "Sulfate/other" are not included in the comparison, as there is no respective category in DeLiAn yet.

The assumptions of the lidar ratio of several aerosol subtypes appear to be in good agreement with the ground-based observations, e.g., for "Clean marine" and "Volcanic ash", mainly due to an overlap in the data sources (Kim et al., 2018). Other aerosol subtype categories such as "Clean continental", "Polluted dust", "Elevated smoke" and "Dusty marine" agree well
within the allowed variability, which is rather large for the CALIPSO data. The large variability of the assigned lidar ratio values to the aforementioned aerosol subtypes can potentially lead to large uncertainties in the retrieved extinction profiles. Kanitz et al. (2014a) had shown that for the version 3 CALIPSO data, the surface-dependent aerosol typing was not allowing for a correct classification of marine aerosol over land, and recently Ansmann et al. (2021) showed that stratospheric smoke

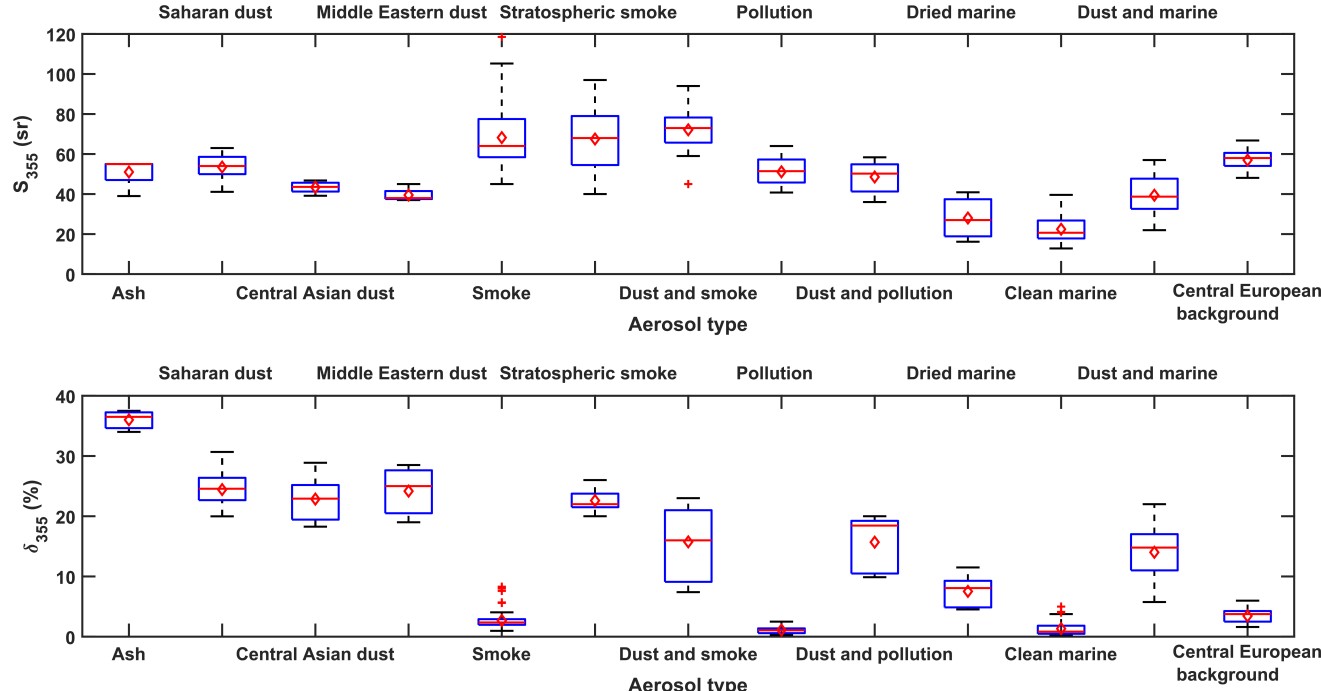

**Figure 5.** Statistics of the lidar ratio (top) and the particle linear depolarization ratio (bottom) at 355 nm for the 13 aerosol categories. The minimum and maximum values are indicated by the lower and upper whisker, median and mean values by the red lines and rhombuses, respectively. The lower part of each box indicates the 25% percentile and the upper part the 75% percentile. Red crosses represent the outliers.

layers are misclassified as sulfate aerosol layers (version 4 CALIPSO data). The present collection could be therefore utilized to
reduce the aerosol-subtype-related variability linked to the 532 nm lidar ratio. Three CALIPSO subtypes namely "Dust", "Polluted continental/smoke" and "Smoke" (stratospheric) seem to be problematic with respect to the present ground-based data collection. "Dust" has a lidar ratio of $44 \pm 9$ sr, which is lower than what we observed for Saharan dust and higher than Central Asian and Middle Eastern dust. An increase of the lidar ratio variability of the specific aerosol subtype or a geographical-specific constraint would be therefore suggested to cover all the observations. The "Polluted continental/smoke" subtype has
the highest variability and the nomenclature itself indicates that several aerosol types can be assigned under that category. With the current allowed variability, this category includes both the Smoke and Pollution categories of the present study, and is therefore correct. However, we would suggest the splitting of this subtype to allow a better discrimination in the radiative effects. Finally, the stratospheric smoke observations (see Tab. 1) suggest that the 532 nm lidar ratio of stratospheric smoke is higher than the one considered in the CALIPSO version 4 "Smoke" subtype (stratospheric).



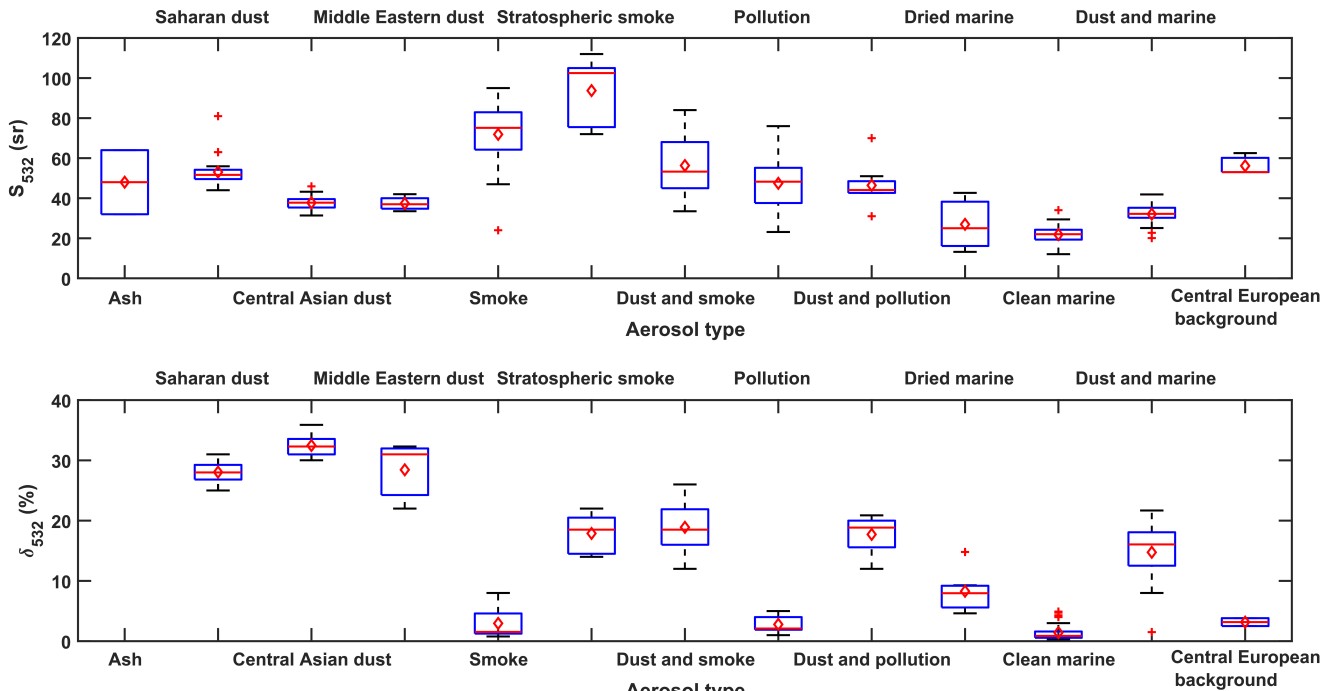

**Figure 6.** Same as in Fig. 5, but for 532 nm.

## 4 Conclusions and outlook

A collection of intensive optical properties (DeLiAn) for various aerosol types based mainly on ground-based lidar observations was presented in this paper. DeLiAn merges measurements from old and more recent campaigns (large temporal coverage) and from various locations (large spatial coverage), and brings together aerosol types that were previously disregarded (e.g., dried marine particles, stratospheric smoke). The optical properties are presented at two wavelengths, 355 and 532 nm, and therefore, can be widely used for aerosol typing purposes, covering spaceborne lidars (CALIOP, ATLID) and ground-based lidar networks (e.g., MPLNET, EARLINET, AD-NET).

The presented statistics cover the most frequently observed aerosol types i.e., smoke, marine, pollution, dust and complex mixtures that they create as well as occasionally observed aerosol types e.g., dried marine particles. Such statistics can have multiple usages and are needed for aerosol classification purposes and aerosol typing schemes since they can be utilized as thresholds separating the different aerosol categories, or as a priori information. DeLiAn has been used for the development of both HETEAC (Wandinger et al., 2016a, Wandinger et al. in preparation) and HETEAC-Flex, a novel aerosol typing approach (Floutsi et al., 2019, Floutsi et al., in preparation), which shall homogenize aerosol typing from EarthCARE but also ground-based lidar systems with different capabilities and it will serve as a ground-based validation scheme for EarthCARE. The data collection of the different intensive optical properties can also be used for further improvement of already-existing aerosol

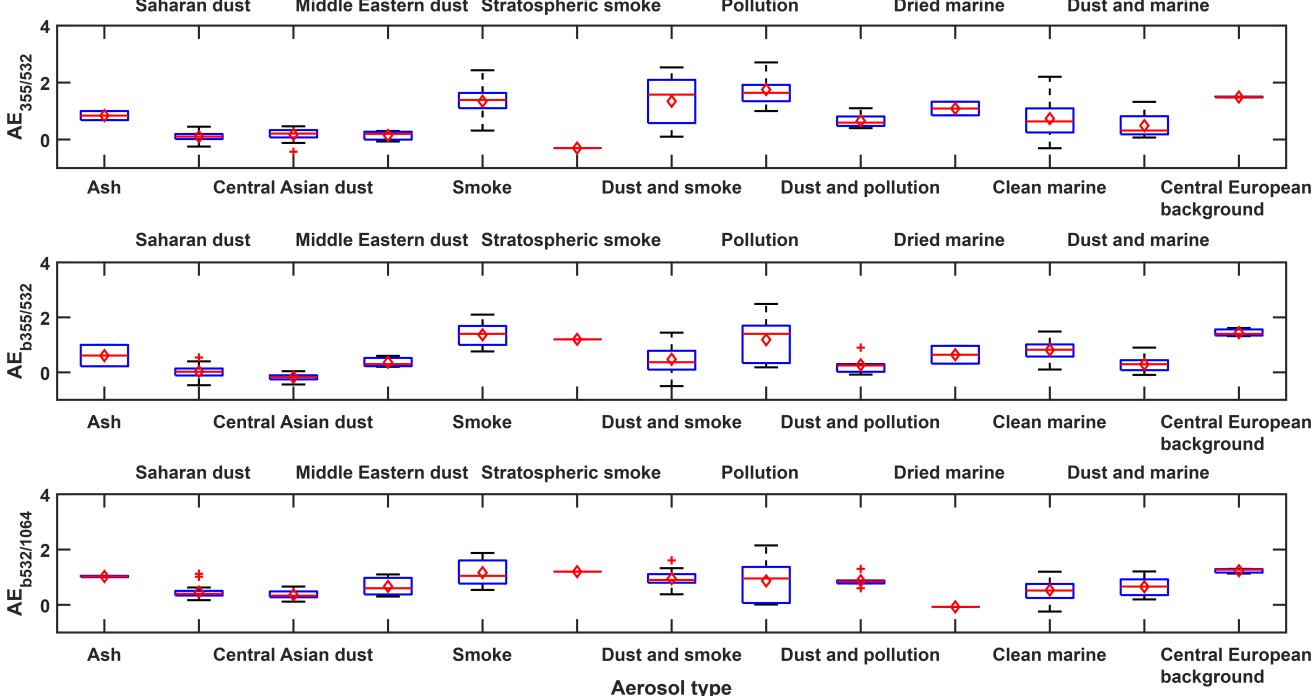

**Figure 7.** Statistics of the extinction-related (top), the 355/532 nm backscatter-related (middle) and the 532/1064 nm backscatter-related (bottom) Ångström exponents for the 13 aerosol categories.

classification schemes, for instance as shown in the case of CALIPSO, where more accurate lidar ratios can lead to better extinction retrievals. Thus, DeLiAn could be considered by the MIRA (Models, In situ, and Remote sensing of Aerosols) working group in their efforts for the creation of an updated CALIPSO lidar ratio climatology (version 5; Schuster et al., 2022).

The presented data collection is only the beginning of a bigger effort for creating an aerosol climatology of intensive optical properties based on ground-based lidar measurements. Measurements from other permanent PollyNET stations are planned to be incorporated in DeLiAn to include geographical regions with interesting aerosol conditions which are currently underrepresented. For instance, measurements from the Eastern Mediterranean region would significantly enrich many aerosol categories such as marine, pollution, dust and their complex mixtures. Heese et al. (2022) recently presented a 2-year long lidar dataset from the coastal city of Haifa, Israel. It was found that most of the observed aerosol layers were aerosol mixtures. Even though a seasonal air-mass source attribution was performed by the authors using TRACE (Radenz et al., 2021b), a more detailed case-by-case air-mass source attribution needs to be performed before the data are included to the data collection.

Data from recent measurement campaigns are also planned to be included in DeLiAn, after data collection and processing is complete. Since the beginning of the ASKOS experiment (Amiridis and the ASKOS team, 2022) in the summer of 2021, a Polly$^{\mathrm{XT}}$ Raman lidar has been fully operational in Mindelo, Cape Verde. Clean marine, mineral dust as well as complex



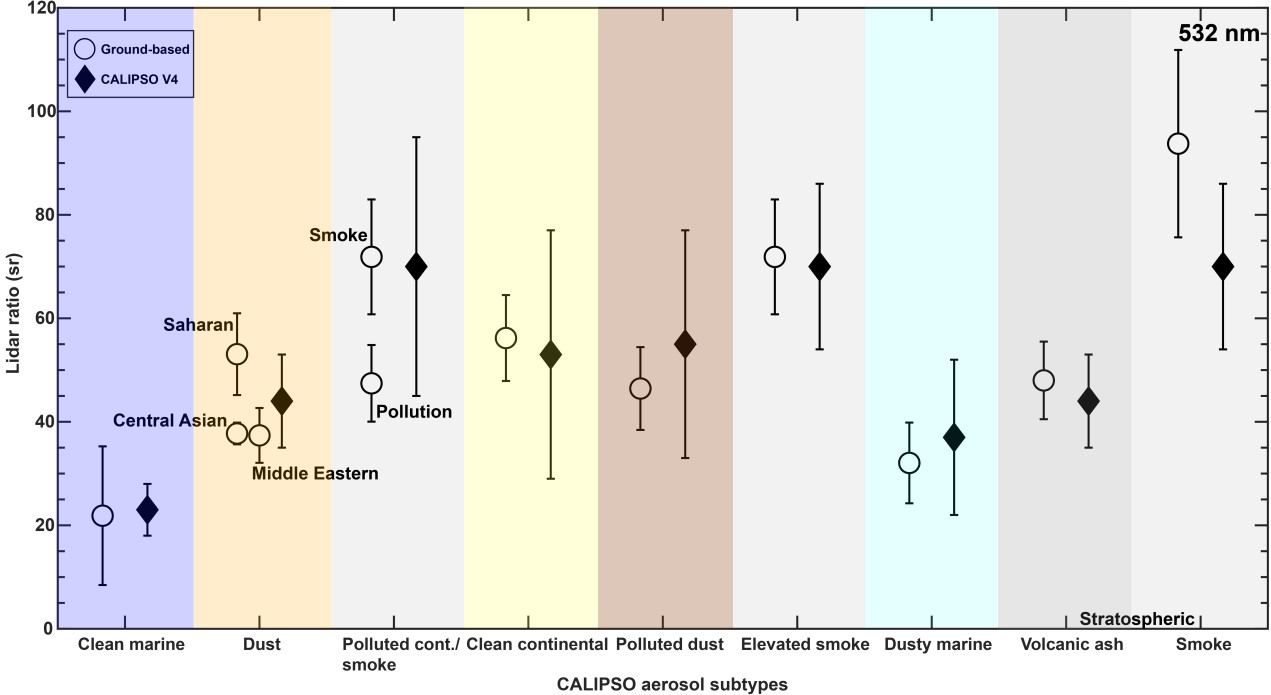

**Figure 8.** Comparison of the 532 nm lidar ratio as derived from ground-based lidars (white circles; this study) and from the CALIPSO aerosol subtypes (black rhombuses; Kim et al., 2018).

510   mixtures that they form have been observed regularly. Dried marine particles have also been observed often, typically on the top of the marine boundary layer.

In a recent study, Haarig et al. (2022) provided the first-ever lidar measurements of the lidar ratio and particle linear depolarization ratio for desert dust particles at all three lidar wavelengths (355, 532 and 1064 nm). As these measurements become more and more available, DeLiAn ought to be updated accordingly, since spectrally resolved information including

515   the 1064 nm, is important for aerosol typing purposes.

*Data availability.* The data collection is planned to be published to Zenodo after the reviewing process is complete.



## Appendix A:  2D spaces of intensive optical properties

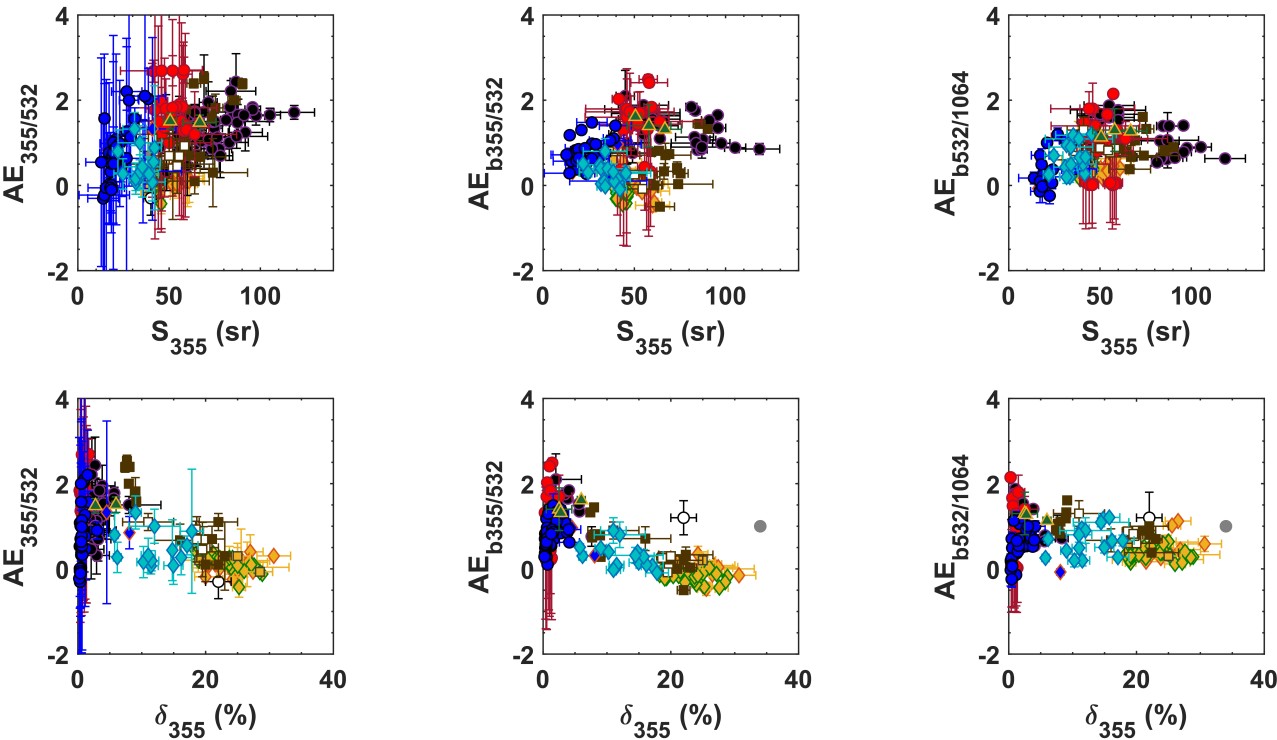

**Figure A1.** 355 nm lidar ratio ($S$) and particle linear depolarization ratio ($\delta$) against the extinction- ($AE_{355/532}$) and backscatter-related ($AE_{b355/532}$) Ångström exponent. The figure legend is the same as in Fig. 2.





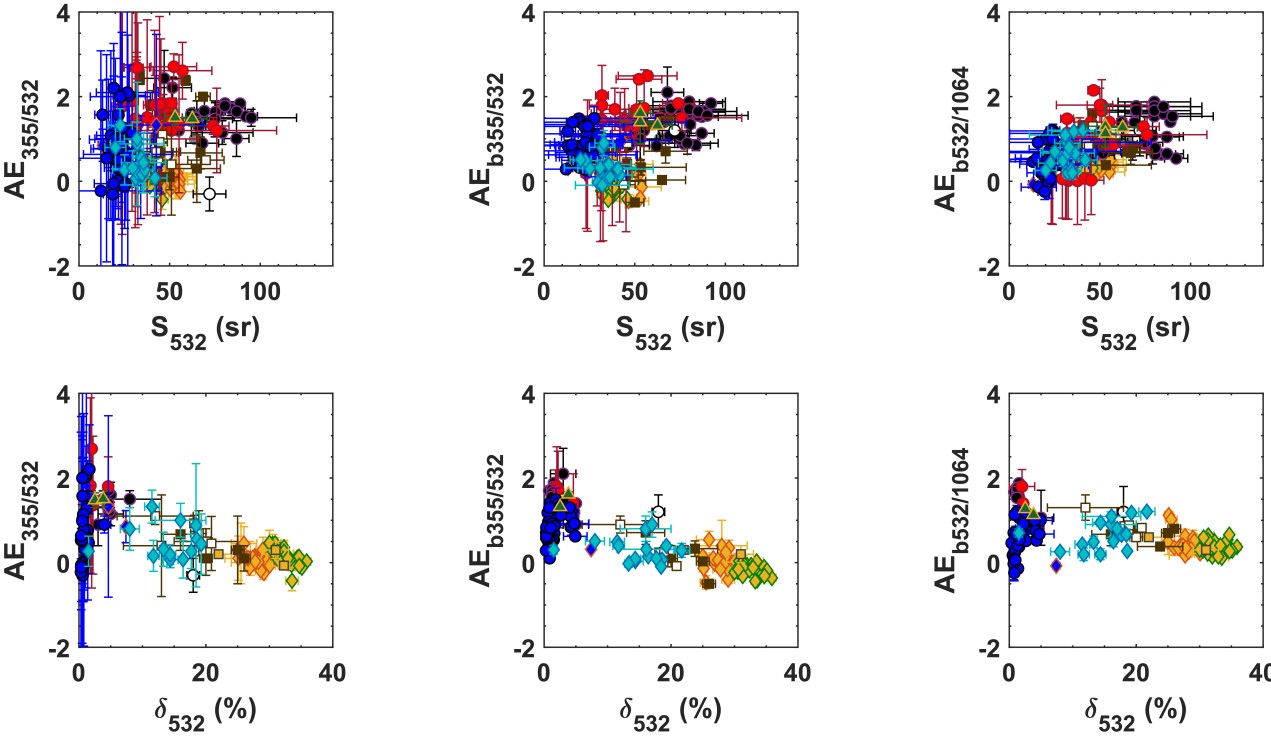

**Figure A2.** Same as Fig. A2, but for the 532 nm lidar ratio ($S$) and particle linear depolarization ratio ($\delta$).

## Appendix B:  Number of data samples per aerosol category

The total number of samples per aerosol category and per intensive property that were considered for the statistics presented in

520     Sec. 3.2 are shown in Tab. B1.





**Table B1.** Number of samples per aerosol category.

| Aerosol type | Total | $S_{355}$ | $\delta_{355}$ | $S_{532}$ | $\delta_{532}$ | $AE_{355/532}$ | $AE_{b355/532}$ | $AE_{b532/1064}$ |
|---|---|---|---|---|---|---|---|---|
| Ash | 4 | 4 | 3 | 2 | - | 2 | 2 | 2 |
| Saharan dust | 30 | 29 | 27 | 26 | 29 | 22 | 19 | 19 |
| Central Asian dust | 23 | 23 | 23 | 23 | 23 | 23 | 23 | 23 |
| Middle Eastern dust | 4 | 4 | 3 | 4 | 3 | 3 | 3 | 3 |
| Smoke | 71 | 70 | 63 | 35 | 19 | 58 | 34 | 29 |
| Stratospheric smoke | 8 | 8 | 5 | 8 | 8 | 1 | 1 | 1 |
| Dust and smoke | 25 | 25 | 25 | 19 | 12 | 13 | 14 | 14 |
| Pollution | 42 | 26 | 20 | 42 | 16 | 25 | 27 | 27 |
| Dust and pollution | 15 | 15 | 13 | 8 | 8 | 8 | 6 | 6 |
| Dried marine | 7 | 3 | 7 | 3 | 7 | 2 | 2 | 1 |
| Clean marine | 52 | 38 | 52 | 48 | 49 | 33 | 48 | 30 |
| Dust and marine | 21 | 20 | 21 | 16 | 16 | 13 | 16 | 16 |
| Central European background | 14 | 14 | 14 | 3 | 2 | 2 | 3 | 3 |





*Author contributions.* The paper was conceptualized and written by AAF and HB. AAF collected, visualized the data and drafted the manuscript. UW provided guidance throughout the study. DA, JH and SFA were involved in the CADEX campaign, AA in the SAMUM–1 and –2 campaigns, PS, MR and BB in the DACAPO-PESO campaign, MH in SALTRACE campaign, and HB, EG and MK in the EUCAARI campaign. MH and REM were involved in the CyCARE and A-LIFE campaigns, EM and VA in the BACCHUS, CyCARE, A-LIFE and PRE-TECT campaigns, AG in the A-LIFE abd PRE-TECT campaigns. MF and MK were responsible for the lidar observations in the United Arab Emirates. TK, SB, KO and MR were involved in Polarstern cruises. Instruments onboard Polarstern have been regularly taken care of by RE, HB and ZY. ISS, LJ, DB and HB are responsible for the PollyNET/EARLINET stations mentioned in this study. DA, BH, AS and RE have continuously contributed to the development and upgrade of Polly$^{\text{XT}}$ lidar systems.

*Competing interests.* UW and VA are members of the editorial board of Atmospheric Measurement Techniques. The authors have no further conflict of interest to declare.

*Acknowledgements.* The authors acknowledge support through the following projects and research programs:

– ACTRIS under grant agreement no. 262 254 of the European Union Seventh Framework Programme (FP7/2007–2013)

– ACTRIS-2 under grant agreement no. 654109 from the European Union's Horizon 2020 research and innovation programme

– EUCAARI funded by the European Union Sixth Framework Programme (FP6) under grant no. 036 833-2

– CADEX funded by the German Federal Ministry of Education and Research (BMBF) under the grant no. 01DK14014

– the Gottfried Willhelm Leibniz Association (OCEANET project in the framework of PAKT)

– BEYOND (funded under: FP7-REGPOT-2012-2013-1) under grant agreement no. 316210.

– "Megacities-Megachallenge – Informal Dynamics of Global Change" (SPP 1233) funded by the German Research Foundation (DFG)

– Foundation of Science and Technology of Poland (FNiTP) Grant no. 519/FNITP/115/2010

– National Science Centre of Poland (NCN, DAINA-2) Grant no. 2020/38/L/ST10/00480

– Polarstern expeditions ANT-XXVI/1, ANT-XXVI/4, ANT-XXVII/1, AWI_PS75_00, AWI_PS77_00, AWI_PS81_00, AWI_PS83_00, AWI_PS95_00, AWI_PS98_00, AWI_PS122_00 and MOSAiC20192020

– SAMUM funded by the Deutsche Forschungsgemeinschaft (DFG) under grant number FOR 539.

– BACCHUS funded by the European Union's 7th Framework Programme (FP7/2007-2013) under grant agreement no. 603445.

– A–LIFE funded by the European Research Council (grant no. 640458).

– EVAA funded by the German Federal Ministry for Economic Affairs and Energy (BMWi) under grant no. 50EE1721C.

– European Research Council (ERC) under the European Community's Horizon 2020 research and innovation framework programme — ERC grant agreement no. 725698 (D-TECT)

– PANhellenic Geophysical Observatory of Antikythera (PANGEA) of the National Observatory of Athens, Greece



550    – "EXCELSIOR": ERATOSTHENES: EXcellence Research Centre for Earth Surveillance and Space-Based Monitoring of the Environment H2020 Widespread Teaming project (www.excelsior2020.eu), funded from the European Union's Horizon 2020 research and innovation programme under Grant Agreement No. 857510, from the Government of the Republic of Cyprus through the Directorate General for the European Programmes, Coordination and Development and the Cyprus University of Technology

   – PoLiCyTa project: PollyXT-CYP funded by the German Federal Ministry of Education and Research (BMBF)

555    – The National Fund for Scientific and Technological Development of Chile, FONDECYT, through grant agreement No. 11181335.

Furthermore, the authors would like to acknowledge and thank all the scientists and technical personnel involved in the realisation of all the measurement campaigns and maintenance of the lidar stations, the RV Meteor team for their support during the M96 cruise, the Alfred Wegener Institute and the RV Polarstern crew during the ship cruises. Many improvements, both, in terms of hard and software were triggered by the fruitful discussions and network activities within EARLINET and ACTRIS.



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
