# Peer review of "DeLiAn – a growing collection of depolarization ratio, lidar ratio and Ångström exponent for different aerosol types and mixtures from ground-based lidar observations"

_Atmospheric Measurement Techniques, 2022_

## Referee Comment (RC1)

The paper by Floutsi et al. presents measurements of three key intensive parameters derived from lidar measurements at several ground-based stations. The paper is important because it compiles measurements of different types of aerosols by ground-based lidar that can be used to provide a universal basis for classifying aerosols and thus improving the retrieval of other parameters such as extinction from lidar measurements. These are particularly useful for determining the aerosol type so that the extinction can be calculated from backscatter measurements that typically need to assume a lidar ratio based on the type of aerosol. The identification of the aerosol type is important for radiation since the radiative transfer depends on the extinction. Though not acknowledged in the paper, aerosol type is important for air quality since the impact on human health depends on the composition of the particles ingested. The paper is well written and easy to follow though I found several areas that could benefit from some clarification and in some cases errors and typos which I will try to address below.

The abstract and introduction are generally well written and comprehensive with the following minor typos
Line 11 - emitted smoke into the stratosphere showing  significantly different optical properties

Line 13 - The paper contains the  most up-to-date comprehensive…

Line 70 "automized" is not an English word. Please write "automated" instead and replace "automized" throughout the document

Line 102 – The HETEAC paper by Wandinger et al is now in AMTD and is beyond "in preparation". Please update citation
Line 117 different locations  over many years.

Line 157 Does the statement "The near-range telescope allows the detection of scattered light (at 355, 387, 532 and 607 nm) from an altitude of around 60–80 m above ground level (AGL)" refer to the overlap height between the laser beam and the receiver field of view of a lidar system, the so-called overlap distance? Please clarify.
Line 158 - Is the maximum height the same for all wavelengths?
Line 164 – Please define the particle linear depolarization using equations to avoid ambiguity
Line 174 – You use Tab. instead of Table. Please use Table throughout the paper.

Table 1. For each aerosol type please specify the values obtained by each measurement individually. This will help the reader to understand the variability between different measurements for each type of aerosol. For example, for ash provide the values obtained Groß et al. (2012), Sicard et al. (2012), Kanitz (2012) separately along with the number of measurements that were used to calculate the respective mean values and standard deviations.

Please present an overview of the types before presenting Table 1. For example how do you define Ash – is it only the silicon quartz mineral content of volcanic eruptions or does it also include sulfuric acid droplets? The composition determines the aerosol properties. While I understand it is difficult to know the composition precisely, you can offer a theory based on the intensive properties and the location of the eruption. This is also why it is important to itemize Table 1 provide the location and campaign for each of the measurements

Also discuss why the intensive properties are invariant with wavelength for smoke but not for stratospheric smoke. Is this true for all campaigns? Can you offer a theory why?
Why is the 532 nm lidar ratio for stratospheric smoke so high?

In Table 2 what are the ± values. As in Table 1, please provide values from individual campaigns for each type/wavelength . Also provide an equation definition AE

Figures 2 and 3 are interesting and do have quite a bit of utility but not in the current form. This is because it's difficult to extract quantitative information from the figures. If I was to use this figure to develop an aerosol typing algorithm that uses the intensive properties presented, I could not because the figure is quite busy. I suggest finding a way to present these results quantitatively. Also, the figures do not present which is the better measure to use for typing the different aerosol types. I would imagine the 532 nm-based relations (Fig 3) are better for the larger particles such as dust and they sold whereas the 355 nm-based relations (Fig 2) are better for fine particles such as smoke and pollution. However, this is just speculation on my part and can be easily verified or discounted by a quantitative measure of for example how wide are the clusters as denoted by the standard deviation oh how far apart are they median values mean values of the different types depicted.

Line 340 – Please present the frequency of the measurements instead writing "…were rare at the time …"

Line 344 – You mean data base (instead of data basis)?

Line 361 – Provide a theory why stratospheric smoke has high depolarization – speculation that can be verified or discounted by others is useful to move these studies forward.

Line 369 – You write "This is a significant finding, as aerosol particles in the stratosphere were usually attributed to volcanic origin (or e.g., generically classified as "stratospheric features" This is no longer true in the latest version of CALIPSO (see Tackett et al in AMTD - https://amt.copernicus.org/preprints/amt-2022-289/)
If you want to keep this sentence, at least qualify it by writing "depolarizing aerosol particles in the stratosphere"

Line 380 – What is the frequency of relative humidity less the 45% in the marine environment so that the reader can appreciate the probability of crystallized seasalt?

What is the optical depth of the Central European Background aerosol? Is the extinction so low that the effect of a higher lidar ratio is not significant because the impact on radiation is very low? If the optical depths of these background layers are consistently lower than 0.05 then we may not need to pay so much attention to them.

Please use the latest CALIPSO publications for your comparisons with CALIPSO. In particular Tackett et al. above can be a great resource.

Line 476 To get a better appreciation of the radiative effects of different aerosol types and subtypes it might help to look at variabilities in the single scattering albedo and asymmetry parameter in addition to the extinction properties.

---

## Author Comment (AC1)

Dear Ali Omar,

First, we would like to thank you for your efforts and valuable feedback, which helped us improve the quality of the paper. In the following, we address your major comments (shown in grey) point by point, with our response formatted in black. Text additions or alterations to the manuscript are shown in blue. Minor comments (such as typos etc.) have been corrected in the revised manuscript, but are not explicitly addressed here.

▷ The paper by Floutsi et al. presents measurements of three key intensive parameters derived from lidar measurements at several ground-based stations. The paper is important because it compiles measurements of different types of aerosols by ground-based lidar that can be used to provide a universal basis for classifying aerosols and thus improving the retrieval of other parameters such as extinction from lidar measurements. These are particularly useful for determining the aerosol type so that the extinction can be calculated from backscatter measurements that typically need to assume a lidar ratio based on the type of aerosol. The identification of the aerosol type is important for radiation since the radiative transfer depends on the extinction. Though not acknowledged in the paper, aerosol type is important for air quality since the impact on human health depends on the composition of the particles ingested. The paper is well written and easy to follow though I found several areas that could benefit from some clarification and in some cases errors and typos which I will try to address below.

As well noticed, in the paper we had not acknowledged the impacts of aerosols on human health. We have included the following statement (now in lines 22-23): "Apart from the impact on the environment, aerosols impact human health as well, and, therefore, aerosol typing is necessary for air quality monitoring and assessment (Fuzzi et al., 2015)."

▷ The abstract and introduction are generally well written and comprehensive with the following minor typos

Line 11 - emitted smoke into the stratosphere showing significant significantly different optical properties

Line 13 - The paper contains the currently most up-to-date comprehensive…

Line 70 "automized" is not an English word. Please write "automated" instead and replace "automized" throughout the document

Line 102 – The HETEAC paper by Wandinger et al is now in AMTD and is beyond "in preparation". Please update citation

Line 117 different locations throughout over many years.

All typos were corrected as suggested and the references have been updated accordingly.

▷ Line 157 Does the statement "The near-range telescope allows the detection of scattered light (at 355, 387, 532 and 607 nm) from an altitude of around 60–80 m above ground level (AGL)" refer to the overlap height between the laser beam and the receiver field of view of a lidar system, the so-called overlap distance? Please clarify.

Thank you for raising a point that needs further clarification. Indeed, the statement refers to the range-dependent overlap between the laser beam and the receiver field of view, which generally is different for every lidar system. However, the overlap characteristics in most Polly systems do not

change significantly due to the standard/fixed optical setup. In some cases, the overlap might be affected by temperature fluctuations and therefore the lower altitude might vary, as stated in the manuscript.

The text (now in lines 203-205) has been revised to: "A second near-range receiver allows the detection of scattered light (at 355, 387, 532 and 607 nm) from a lower altitude of around 60–80 m above ground level (AGL) due to the laser-beam overlap with the receiving telescope."

▷ Line 158 - Is the maximum height the same for all wavelengths?

The statement in line 158 ("The uppermost detection height for the vertical profiles is around 20 km AGL.") was inaccurate, thank you for bringing this up. The lidar signals of our PollyXT systems are recorded up to 46 km. This is wavelength independent. Nevertheless, depending on the instrument performance (laser power, background light, etc.) the maximum height of useful signal varies, but is usually at least up to 20 km. In rare events (e.g., Australian wildfires), we have observed aerosol layers up to 30 km (Ohneiser et al., 2022). We have rephrased the corresponding statement to (now in lines 205-207): "The lidar signals are recorded up to 46 km, but depending on the instrument performance (laser power, background light, etc.) the maximum height of useful signal varies, usually reaching at least up to 20 km."

▷ Line 164 – Please define the particle linear *depolarization* using equations to avoid ambiguity

Thank you for your comment. To avoid ambiguity we have now added a new section (now in lines 130-175), where we describe the intensive optical properties used for DeLiAn along with the corresponding equations. In particular, the particle linear depolarization ratio is discussed in lines 149-164.

▷ Line 174 – You use Tab. instead of Table. Please use Table throughout the paper.

The whole manuscript has been updated with respect to this comment, thank you.

▷ Table 1. For each aerosol type please specify the values obtained by each measurement individually. This will help the reader to understand the variability between different measurements for each type of aerosol. For example, for ash provide the values obtained Groß et al. (2012), Sicard et al. (2012), Kanitz (2012) separately along with the number of measurements that were used to calculate the respective mean values and standard deviations.

Thank you for this comment. We understand the necessity for clearly presenting each and every measurement used for each aerosol type. However, that would require reporting each measurement for a total of seven parameters (lidar ratio and particle linear depolarization ratio at two wavelengths and the Ångström exponent for three wavelength pairs) along with the respective observational errors. The number of individual measurements that would need to be listed per aerosol category is provided in Table B1 (stated in line 523 of the updated manuscript). Reporting every single measurement would create a very busy table that would eventually lose its functionality. DeLiAn is now publicly available on Zenodo (https://doi.org/10.5281/zenodo.7751752).

▷ Please present an overview of the types before presenting Table 1. For example how do you define Ash – is it only the silicon quartz mineral content of volcanic eruptions or does it also include sulfuric acid droplets? The composition determines the aerosol properties. While I understand it is difficult to know the composition precisely, you can offer a theory based on the intensive properties and the location of the eruption. This is also why it is important to itemize Table 1 provide the location and campaign for each of the measurements

Volcanic eruptions typically produce a mixture of ash particles (such as volcanic glass and minerals) and volcanic gases (such as sulphur dioxide and carbon dioxide) among others (e.g., hydrogen, hydrochloric acid, etc.). The bigger in size particles, due to their mass, typically fall out quite fast while smaller in size particles (such as sulfate aerosol) have a higher residence time in the atmosphere. The sulfate particles form from the precursor gases only after several days. As stated in your comment, it is difficult to know the exact composition without an in-situ chemical analysis, nevertheless, we would like to comment on the measurements that have been selected for DeLiAn. The observations were made in fresh volcanic plumes (a few days after the eruption). All the included measurements have a particle linear depolarization higher than 34% (355 nm). These values can only be associated with mineral or glass particles (non-spherical) and not sulfuric acid droplets or any other hygroscopic species that may attach to the ash particles and make them more spherical. For clarity, we have rephrased the text (now in line 350): "The volcanic ash category contains measurements of fresh mineral particles from the Eyjafjallajökull eruption…". In addition, the location of the measurements is included in the published data collection (https://doi.org/10.5281/zenodo.7751752).

▷ Also discuss why the intensive properties are invariant with wavelength for smoke but not for stratospheric smoke. Is this true for all campaigns? Can you offer a theory why? Why is the 532 nm lidar ratio for stratospheric smoke so high?

This is a great comment, thank you for bringing it up. The negligible wavelength dependence of the lidar ratio is a signature of fresh smoke particles. This has been observed in several field campaigns (e.g., in Amazonia (Baars et al., 2012), in Cabo Verde (Tesche, 2011)). On the contrary, aged smoke exhibits a strong spectral dependency of the backscatter coefficient and at the same time a weak spectral dependency of the extinction coefficient. Consequently, the 355-nm lidar ratio is considerably lower compared to the one at 532 nm (Müller et al., 2005, Ansmann et al., 2009). The typical deviation between the 355 and 532-nm lidar ratio is around 20 sr (Haarig et al., 2018; Ohneiser et al., 2021).

DeLiAn's broader smoke category includes both measurements from fresh and aged smoke. For that reason, we have carefully revised the respective phrasing in the manuscript to include information on the age of the smoke particles (see now lines 395-417).

In the case of stratospheric smoke, the spectral behaviour of the lidar ratio is similar to the one of aged (tropospheric) smoke, i.e., it increases with wavelength.

▷ In Table 2 what are the ± values. As in Table 1, please provide values from individual campaigns for each type/wavelength . Also provide an equation definition AE

The ± values reported in Table 2 correspond to the mean observational error for every aerosol category. For clarity, we have updated the table captions for both Tables 1 and 2. For the individual AE measurements please refer to the published collection (https://doi.org/10.5281/zenodo.7751752). In addition, we have also have described and defined the Ångström exponent (now in lines 165-175).

▷ Figures 2 and 3 are interesting and do have quite a bit of utility but not in the current form. This is because it's difficult to extract quantitative information from the figures. If I was to use this figure to develop an aerosol typing algorithm that uses the intensive properties presented, I could not because the figure is quite busy. I suggest finding a way to present these results quantitatively. Also, the figures do not present which is the better measure to use for typing the different aerosol types. I would imagine the 532 nm-based relations (Fig 3) are better for the larger particles such as dust and they sold whereas the 355 nm-based relations (Fig 2) are better for fine particles such as smoke and pollution. However, this is just speculation on my part and can be easily verified or

discounted by a quantitative measure of for example how wide are the clusters as denoted by the standard deviation oh how far apart are they median values mean values of the different types depicted.

Thank you for your suggestion. We are aware that indeed, Figures 2 and 3 are very busy, especially for the 355-nm, however, the Figures are meant only to be a visual representation of DeLiAn. To be honest, we tried several different ways to visualize the data before deciding to show those two Figures in the manuscript. In this paper, we aim to present DeLiAn and this is just an initial version as stated in the outlook, more observations and more aerosol categories are to be added. In the long term, it will definitely be impossible to find a way to adequately visualize all the information. To obtain quantitative information one can use the data collection itself (https://doi.org/10.5281/zenodo.7751752); it's not necessary at all to extract any data from the Figures.

▷ Line 340 – Please present the frequency of the measurements instead writing "...were rare at the time ..."

With this phrase, we only aim to convey a simple message: that measurements of particle linear depolarization ratio were usually performed at 532 nm and the EUCAARI campaign provided one of the first datasets of particle linear depolarization ratio at 355 nm. Unfortunately, we do not have any quantitative information on the number of measurements performed and, therefore, we cannot report any frequency of measurements here. Nevertheless, we have rephrased the statement (now in lines 397-400) to: "At the time of the campaign, measurements of lidar ratio and particle linear depolarization ratio were usually conducted at 532 nm, while measurements at 355 nm (especially of particle linear depolarization ratio) were only occasionally performed (depicted with faded black/purple circles in Fig. 2)."

▷ Line 344 – You mean data base (instead of data basis)?

Thank you, we meant data collection and it has been rephrased (now in line 405).

▷ Line 361 – Provide a theory why stratospheric smoke has high depolarization – speculation that can be verified or discounted by others is useful to move these studies forward.

We have rephrased the lines 421-425 of the updated manuscript ("In contrast to tropospheric smoke, which contains mainly spherical particles, stratospheric smoke consists of consists of non-spherical soot particles, yielding high depolarization ratios…"). Furthermore, we have included a new statement (now in lines 425-426): "The enhanced depolarization ratios, along with the characteristic lidar ratio wavelength dependence of the stratospheric smoke were recently modeled by Gialitaki et al. (2020)."

▷ Line 369 – You write "This is a significant finding, as aerosol particles in the stratosphere were usually attributed to volcanic origin (or e.g., generically classified as "stratospheric features" This is no longer true in the latest version of CALIPSO (see Tackett et al in AMTD - https://amt.copernicus.org/preprints/amt-2022-289/)
If you want to keep this sentence, at least qualify it by writing "depolarizing aerosol particles in the stratosphere"

Thank you for bringing this up, you are right. We have updated the statement (now in line 432). For clarity, we have also included the following statement (now in lines 433-435): "However, the classification of stratospheric smoke layers during night time appears to be improved after the

introduction of the CALIPSO version 4.5 stratospheric aerosol subtyping algorithm (Tackett et al., 2023)."

▷ Line 380 – What is the frequency of relative humidity less the 45% in the marine environment so that the reader can appreciate the probability of crystallized seasalt?

Thank you for this comment. Observing dried marine aerosol is complex. Based on our PollyXT observations, dried marine aerosol in marine environments can be found usually during the summer months, on the top of the boundary layer, in the transition zone to the dry free troposphere, forming a geometrically thin layer of only a few meters in height (~100 m) (Bohlmann et al., 2018). As stated in the manuscript, measurements of this particular aerosol type are sparse, but the frequency of observing a relative humidity of less than 45% in a marine environment is not known, due to the lack of humidity measurements itself. Typical relative humidity values in the marine boundary layer are around 80% (Haarig et al., 2017). In a recent study (Thomas et al., 2022), the variation of the marine aerosol properties (over the Southern Ocean) was investigated with respect to various meteorological conditions, including relative humidity. Based on CALIPSO vertically resolved aerosol properties, it was shown that the particle linear depolarization ratio is sensitive to relative humidity (see Figure 2 of the respective paper). It was shown that marine aerosol particles in the marine boundary layer exhibit higher particle linear depolarization ratio values when the relative humidity drops below 60%. The vertical structure of the particle linear depolarization ratio below and above 60% is nicely demonstrated in Figure 9.

We have reworked the dried marine section of the manuscript in order to emphasize the rarity and importance of this peculiar aerosol type for aerosol typing applications. To be precise, the following sentence has been updated (now in lines 458-462): "The optical properties of aerosol of marine origin do not show variability with respect to the source, since they are usually observed in environments of high relative humidity (typically 60–80%) and the particles have a spherical shape due to water uptake (Haarig et al., 2017b; Thomas et al., 2022). However, in more rare cases, when …"

▷ What is the optical depth of the Central European Background aerosol? Is the extinction so low that the effect of a higher lidar ratio is not significant because the impact on radiation is very low? If the optical depths of these background layers are consistently lower than 0.05 then we may not need to pay so much attention to them.

Thank you for raising this point. In DeLiAn, the Central European background aerosol measurements come from stations that are located in the designated geographical location during periods when no advection of aerosol takes place and particles are confined within the planetary boundary layer, exhibiting an aerosol optical thickness of less than 0.2. Even though the lidar ratio of the Central European background is higher than the one of Pollution (at both wavelengths), based on the error bars and the low number of samples in the Central European background category (especially at 532 nm, see Table B1), it is safe to assume that the findings are not statistically significant and that more measurements of this specific aerosol category are needed.

We have carefully edited the text describing the Central European background (now in lines 473-483).

▷ Please use the latest CALIPSO publications for your comparisons with CALIPSO. In particular Tackett et al. above can be a great resource.

We have updated Figure 8 according to the values reported in Tackett et al., 2023. In addition, we have revised all the corresponding text in Section 3.2 accordingly.

▷ Line 476 To get a better appreciation of the radiative effects of different aerosol types and subtypes it might help to look at variabilities in the single scattering albedo and asymmetry parameter in addition to the extinction properties.

While this is an excellent idea, the single scattering albedo and the asymmetry parameter are quantities that we cannot directly retrieve with our lidars. However, in Wandinger et al., 2022, in EGUsphere (https://doi.org/10.5194/egusphere-2022-1241) we provide the respective microphysical model to estimate the radiative properties of different aerosol types based on DeLiAn.

In addition to the comments addressed, we would like to inform you about the following changes in the manuscript:

- New section 2.1 now describes the intensive optical properties
- Section 2.2.3: parts of the description have been updated and others omitted for clarity
- Figure 1: updated background Earth map
- The name of Cabo Verde has been corrected (was Cape Verde)
- Figures A1 and A2 have been reworked to increase readability
- Data availability statement

---

## Author Comment (AC2)

Dear reviewer,

We would like to thank you for carefully reading the manuscript and providing useful feedback to improve the paper. In the following, we address the major comments (shown in grey) point by point, with our response formatted in black. Text additions or alterations to the manuscript are shown in blue.

▷ Although such information might be available in the publications referenced, the authors should provide a comment, how they define from the measurement conditions an aerosol type as pure. Do they consider only the location of the site or they use also other tools such as trajectories or models?

Thank you for pointing out the need for further clarification. When we refer to pure aerosol types, we refer to the observation of single aerosol types such as marine, smoke, pollution, or dust. Therefore, an aerosol mixture can never be considered as a pure aerosol type.

Even though primarily the optical properties are the major criterion for typing, several parameters are taken into account when it comes to aerosol characterization, including the meteorology (backward trajectories), location and altitude of the aerosol plume, advection, connection with e.g., a big event such as a volcanic eruption or wildfires, etc.

We have included a new statement in the revised version of the manuscript (now lines 226-232): "Along with the determination of the intensive optical properties, which play a crucial role in the categorization of the observed particles, other tools such as e.g., backtrajectories are also widely considered. Trajectory and particle dispersion models (e.g., HYSPLIT, FLEXPART; Stein et al., 2015; Pisso et al., 2019) provide valuable information about the source, the distance traveled and the destination of an air-mass for a specific transport time (simulation performed either backward or forward in time). Recently, an automated air-mass source attribution tool, which combines backward trajectories (or particle positions from a dispersion model) with geographical information (land cover classification), TRACE (Radenz et al., 2021b), was developed at TROPOS."

▷ Do the authors consider the ageing of the observed aerosols as a parameter for the typing (this was found in previous studies to be crucial especially for smoke)? A relevant comment should be added in the discussion.

This is a very good point. In the current version of DeLiAn we do not consider further classification based on the aging of the aerosol. For instance, the broader smoke category includes both measurements from fresh and aged smoke. However, we have revised the "Smoke" paragraph carefully and now it provides more information with respect to the age of the smoke particles (see now lines 395-417).

▷ It is confusing, as written, how the authors distinguish "pollution" type and "central European background". More or less for both categories they use measurements from the same stations. They should provide a comment, why in certain cases they consider an observation as representative for pollution and why as background.

Thank you for your comment. Indeed, the "Pollution" and "Central European background" aerosol categories are optically similar and the measurements "share" stations. For instance, in Leipzig, we observed both aerosol types. However, a measurement is considered as representative for Central European background when the following criteria are met: station located in the indicated geographical location, no advection takes place and the aerosol layers must be confined to the

planetary boundary layer and exhibit an aerosol optical thickness of less than 0.2. For clarity, we have updated the text "Central European background" category (now in lines 476-480): "An aerosol layer must follow certain criteria to be categorized as Central European background aerosol, which include the absence of advection of aerosol, the confinement of the particles within the planetary boundary layer and an optical thickness of less than 0.2. In this way, both Central European background and Pollution categories can be separated, even though they both contain mainly aerosol of anthropogenic origin."

▷ The authors group separately mixtures of different aerosol types, especially dust with smoke, dust with pollution and dust with marine. They should provide more details how they define an aerosol scene as a mixture. To my understanding they average all relevant scenes in order to provide a representative value for a certain mixture. Does the mixing ratio of the pure types involved play a role in the typing and do the authors claim that this ratio is not significantly different from location to location?

Thank you for bringing this up. Indeed, lofted layers carrying desert dust are subject to long-range transport and, therefore, mixtures of dust with other aerosol types are dominating.

An aerosol layer is considered a mixture first and foremost based on the intensive optical properties and the information known from the literature. However, this is not the only source of information that helps the correct assignment of an observation to an aerosol type. Tools such as trajectory and aerosol dispersion models are very effective in the correct characterization of the observed particles as they provide information of the source, altitude and distance that an air mass travelled prior to the observation. For clarity, we have updated the manuscript (now in lines 485-488): "Apart from pure aerosol types, aerosol mixtures of dust particles with smoke, pollution and marine particles have been considered in DeLiAn. The determination of the main aerosol types present in an aerosol mixture (performed by the authors of the respective studies) was based on combined information on the intensive optical properties of the aerosol layers and air-mass analysis with the help of trajectory or particle dispersion modelling." With respect to DeLiAn, indeed, we average all the available known mixtures to provide representative values for the intensive properties (Table 1). The individual observations are visualized in Figures 2 and 3 and the data collection is publicly available via Zenodo (https://doi.org/10.5281/zenodo.7751752). The mixing ratio of the pure types, especially the dust contribution, plays a role in the observed aerosol properties. For instance, dust and marine mixtures with higher contributions of dust exhibit significantly higher values of particle linear depolarization ratio compared to those with lower dust contributions (note the wide spread of the dust and marine category in Figures 2 and 3), regardless of the observation location. This effect is also nicely visualized in Figure 7 of Wandinger et al., 2022, in EGUsphere (https://doi.org/10.5194/egusphere-2022-1241), where an aerosol microphysical model is being described based on the DeLiAn observations.

In addition to the comments addressed, we would like to inform you about the following changes in the manuscript:

- New section 2.1 now describes the intensive optical properties
- Section 2.2.3: parts of the description have been updated and others omitted for clarity
- Figure 1: updated background Earth map
- The name of Cabo Verde has been corrected (was Cape Verde)
- Figures A1 and A2 have been reworked to increase readability
- Data availability statement